DOI: 10.1038/s41467-017-00936-3　　**OPEN**

# Distance-dependent inhibition facilitates focality of gamma oscillations in the dentate gyrus

Michael Strüber [1,2], Jonas-Frederic Sauer[1], Peter Jonas[3] & Marlene Bartos [1]

Gamma oscillations (30–150 Hz) in neuronal networks are associated with the processing and recall of information. We measured local field potentials in the dentate gyrus of freely moving mice and found that gamma activity occurs in bursts, which are highly heterogeneous in their spatial extensions, ranging from focal to global coherent events. Synaptic communication among perisomatic-inhibitory interneurons (PIIs) is thought to play an important role in the generation of hippocampal gamma patterns. However, how neuronal circuits can generate synchronous oscillations at different spatial scales is unknown. We analyzed paired recordings in dentate gyrus slices and show that synaptic signaling at interneuron-interneuron synapses is distance dependent. Synaptic strength declines whereas the duration of inhibitory signals increases with axonal distance among interconnected PIIs. Using neuronal network modeling, we show that distance-dependent inhibition generates multiple highly synchronous focal gamma bursts allowing the network to process complex inputs in parallel in flexibly organized neuronal centers.

[1] Physiologisches Institut I, Systemic and Cellular Neurophysiology, Albert-Ludwigs-Universität Freiburg, Hermann-Herder-Straße 7, 79104 Freiburg, Germany. [2] Spemann Graduate School of Biology and Medicine (SGBM), Albert-Ludwigs-Universität Freiburg, 79104 Freiburg, Germany. [3] IST Austria (Institute of Science and Technology Austria), Am Campus 1, 3400 Klosterneuburg, Austria. Correspondence and requests for materials should be addressed to M.Süb. (email: michael_strueber@hotmail.com) or to M.B. (email: marlene.bartos@physiologie.uni-freiburg.de)

Gamma frequency oscillations are important for the processing and storing of information in cortical networks. They are proposed to act as reference signals for discrete representation, segmentation, and selection of information[1–3]. Gamma oscillations have been further hypothesized to support the binding of flexible cell assemblies defined as neuron groups that are transiently synchronized to transfer and store information to control behavior[1, 4, 5]. Simultaneous local field potential (LFP) recordings from different hippocampal areas have shown that gamma rhythms can be classified in low- (30–75 Hz) and high- (75–150 Hz) frequency oscillations[6–10]. In CA1, both frequency bands specifically occur during different phases of hippocampal coding and seem to route information flow originating from distinct brain areas to CA1[6, 8, 9]. Low-gamma activity emerges in the entire hippocampal network during spatial explorative behavior[7], thereby producing a global temporal "context" for encoding of information[1]. Recent investigations propose the additional emergence of focal high-frequency gamma activity in the hippocampus and the prefrontal cortex during conditions of enhanced multisensory information processing and in association with action selection[11, 12]. The burst-like appearance[11] and the sharp decline of gamma power as a function of distance between neighboring recording sites[2, 10, 13, 14] indicate that high-frequency gamma centers are generated by local synaptic interactions within active "microcircuits"[15–18]. Thus, in vivo, rapid network oscillations can be spatially restricted[19] and may emerge during particular behavioral demands[11]. However, the cellular and synaptic mechanisms underlying the generation of focal gamma bursts remains unknown.

Several lines of evidence indicate that GABAergic interneurons (INs), specifically parvalbumin (PV)-expressing perisomatic-inhibitory interneurons (PIIs), are critical for the generation of gamma activity[20–22]. PIIs discharge at high frequencies tightly phase-locked to gamma cycles in vivo[23]. They entrain the activity of their postsynaptic partners and synchronize spike discharges in principal cell assemblies[22, 24]. In the hippocampus, they possess extensive axonal arbors, which innervate a great number of target cells dispersed over a large volume[25]. This morphological phenotype seems optimal to support large scale network synchronization at low-gamma frequency but less for focal fast network

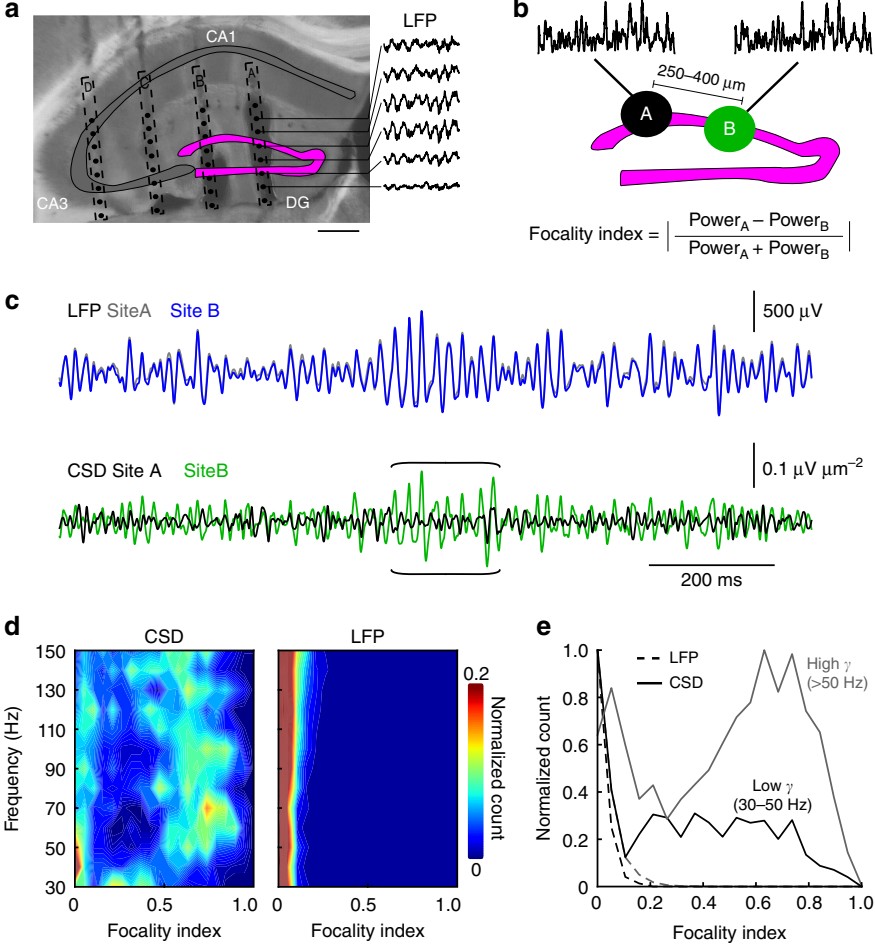

**Fig. 1** Focal gamma bursts emerge in the dentate gyrus of freely moving mice. **a** Electrode tracks of the silicon probe used for LFP recording (shank spacing: 250–400 μm, electrode pitch: 25–100 μm, 8–16 recording sites/shank). Insets on the right show raw LFP traces recorded at the sites indicated. The position of the recording sites was determined post hoc (see "Methods"). Scale bar 400 μm. **b** Schematic illustration of the method used to quantify the focality of gamma oscillations. For each gamma epoch detected on recording sites A and B within the dentate gyrus granule cell layer (gcl) on neighboring shanks, we measured a focality index, which ranges theoretically from 0 (equal gamma power on both recording sites A and B) to 1 (total gamma power located on one of the recording sites A or B). **c** Gamma-filtered (30–150 Hz) LFP (top) and the resulting CSD traces (bottom) of recordings in the gcl at 250 μm shank spacing. Note that the CSD reveals epochs of local gamma activity on one shank (brackets) while no difference in gamma amplitude is apparent when considering LFPs. **d** 2D histogram of focality index as a function of gamma frequency for CSD (left) and LFP (right). High focality indices are observed at frequencies above ~50 Hz in CSD but not LFP traces. In CSD recordings, for frequencies < 50 Hz focality indices are mostly < 0.1. **e** Normalized histograms of focality indices of low gamma (30–50 Hz, black) and high gamma (50–150 Hz, gray) epochs. Data are from N = 4 mice

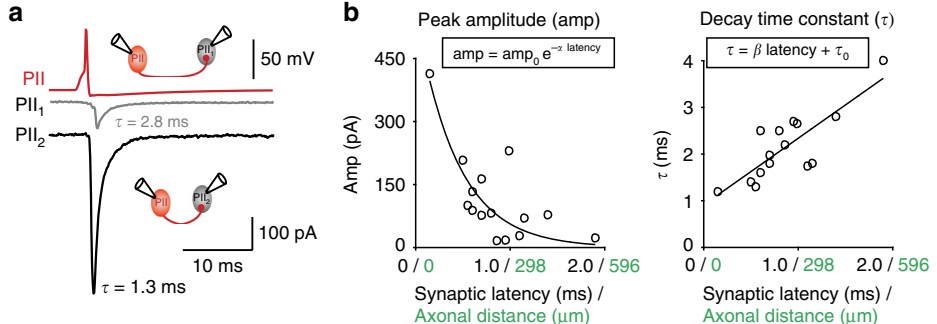

**Fig. 2** Distance-dependent inhibition at PII–PII synapses. **a** Two representative paired PII–PII whole-cell recordings in dentate gyrus slices[36]. Top, presynaptic PII action potential (AP). Bottom, average uIPSCs recorded in two postsynaptic PIIs (gray and black). **b** Peak amplitude (amp, left) and decay time constants ($\tau$, right) of $N = 15$ PII–PII connections plotted against the respective synaptic latency and the calculated axonal distance (distance = 0.29 mm ms$^{-1}$ * latency + 0.01 mm)[26]. Decay time constants were measured by biexponential fits and are given as amplitude-weighted $\tau$[26]. Lines represent exponential and linear fit functions with DD coefficients $\alpha$ and $\beta$ (equations above). Spearman's correlation analysis: amp, $p = 0.01$; $\tau$, $p = 0.002$

oscillations. One solution to this problem might lie in the non-uniformity of PII output signaling, which we recently identified at PII–granule cell (GC) synapses in the rat dentate gyrus[26]. Inhibitory output signaling of single PIIs is strong and rapid to nearby GCs and becomes weaker and slower with axonal as well as inter-somatic distance to remote partners[26]. We termed this form of inhibitory signaling "distance-dependent". However, whether distance-dependent inhibition exists among PIIs remained unknown. Here we demonstrate in freely moving mice that, indeed, the dentate gyrus generates bursts of gamma oscillations, which vary in their spatial range from global to highly focal activity. We then investigate the spatiotemporal profile of mutual inhibition between PIIs in the dentate gyrus and show by re-analyzing whole-cell paired recordings from PIIs in slices of the dentate gyrus that PII–PII synapses are indeed distance-dependent. Our subsequent computational analysis reveals that distance-dependent inhibition supports the emergence of focal gamma bursts. Finally, we demonstrate that this form of synaptic inhibition can support the co-existence of functionally independent focal gamma bursts in hippocampal circuits, which potentially allows for parallel information processing in cortical networks.

## Results

**Focal gamma bursts in dentate gyrus of freely moving mice.** To examine the spatial profile of gamma oscillations in the dentate gyrus of freely moving mice, we recorded LFPs with two-dimensional (2D) silicon probe arrays chronically implanted in the dorsal dentate gyrus (four mice; Fig. 1). To obtain information about the local network activity in the granule cell layer (gcl), we applied current source density (CSD) analysis over individual shanks (Fig. 1a). The recording site situated closest to the gcl was identified by post hoc histology. During unrestricted movement in an open field arena, gamma frequency varied substantially over time at all recording sites. Therefore, we quantified the locality of oscillations over the entire gamma frequency range of 30–150 Hz (10 Hz bandwidth) by first isolating epochs of large gamma power (Fig. 1b; see "Methods")[27]. Next, we determined for all gamma epochs the spatial focality index defined as the inter-shank difference in oscillation power (Fig. 1b; Methods, Equation (2)). The focality index theoretically ranges between 0 and 1, with 0 pointing to no focality and 1 indicating maximal selectivity of gamma power to one recording site. We observed that individual gamma epochs broadly varied in their spatial extent. Interestingly, focality indices of individual gamma epochs were positively correlated with their oscillation frequency (Spearman's $\rho = 0.96$, $p = 3*10^{-7}$). To further investigate the

relationship between frequency and focality, we computed 2D histograms showing the distribution of gamma bursts of different frequencies and spatial spreads in CSD and LFP recordings (Fig. 1d, e). While gamma activity in all frequency ranges appeared to be highly global in LFP recordings, the CSD analysis detected global gamma in the low-gamma range (<50 Hz) and focal gamma concentrated in the higher frequency range (>50 Hz). Thus, dentate gyrus gamma activity patterns vary substantially in frequency and space ranging from global low frequency to focal high frequency oscillations.

**Distance-dependent inhibition between PIIs.** According to one classical theory, hippocampal gamma oscillations arise from mutual interaction between GABAergic cells, in particular, PIIs[20, 28–30]. This seems to be specifically the case for the dentate gyrus, in which neuronal activity stays under tight inhibitory control[20, 28, 31–33] and principal cell activity is extremely sparse[34, 35]. Our finding of highly focal gamma activity in this brain area inspired us to examine the spatiotemporal profile of PII–PII inhibitory signaling. We, therefore, accessed our database and re-analyzed from 15 PII–PII paired whole-cell recordings in the dentate gyrus[36] the relationship between amplitude and decay time constant ($\tau$) of the synaptic signals to the spatiotemporal distance between the synaptic partners (Fig. 2). As a measure for spatiotemporal distance, we used the synaptic latency, which had been shown to linearly correlate with the axonal distance between two connected cells[26]. Similar to our findings for PII–GC connections[26], the amplitude of unitary inhibitory postsynaptic currents (uIPSCs) in PII–PII pairs correlated inversely with the axonal distance between pre- and postsynaptic cell (Spearman's $\rho = -0.64$, $p = 0.010$; Fig. 2b). We fitted this relationship with a negative exponential function decaying with the distance-dependence (DD) coefficient $\alpha = 2.3$ ms$^{-1}$. Moreover, $\tau$ of uIPSCs linearly increased as a function of axonal distance with a DD coefficient $\beta = 1.4$ (Spearman's $\rho = 0.73$, $p = 0.002$; Fig. 2b). To further examine the observed DD of mutual perisomatic inhibition, we intracellularly labeled individual PIIs in hippocampal slices and microscopically analyzed the number and location of their synaptic outputs on immunofluorescently stained postsynaptic target PIIs (Supplementary Fig. 1). Interestingly, the number of putative synaptic contacts declined with the intersomatic distance between two PIIs, while their perisomatic location remained the same (Supplementary Fig. 1). Thus, inhibitory signaling among PIIs is distance dependent. It is strong and rapid at closely spaced cells and becomes weaker and longer lasting at more distant target PIIs, implying a distance-dependent rule of inhibitory PII output signaling in the dentate gyrus of the hippocampus.

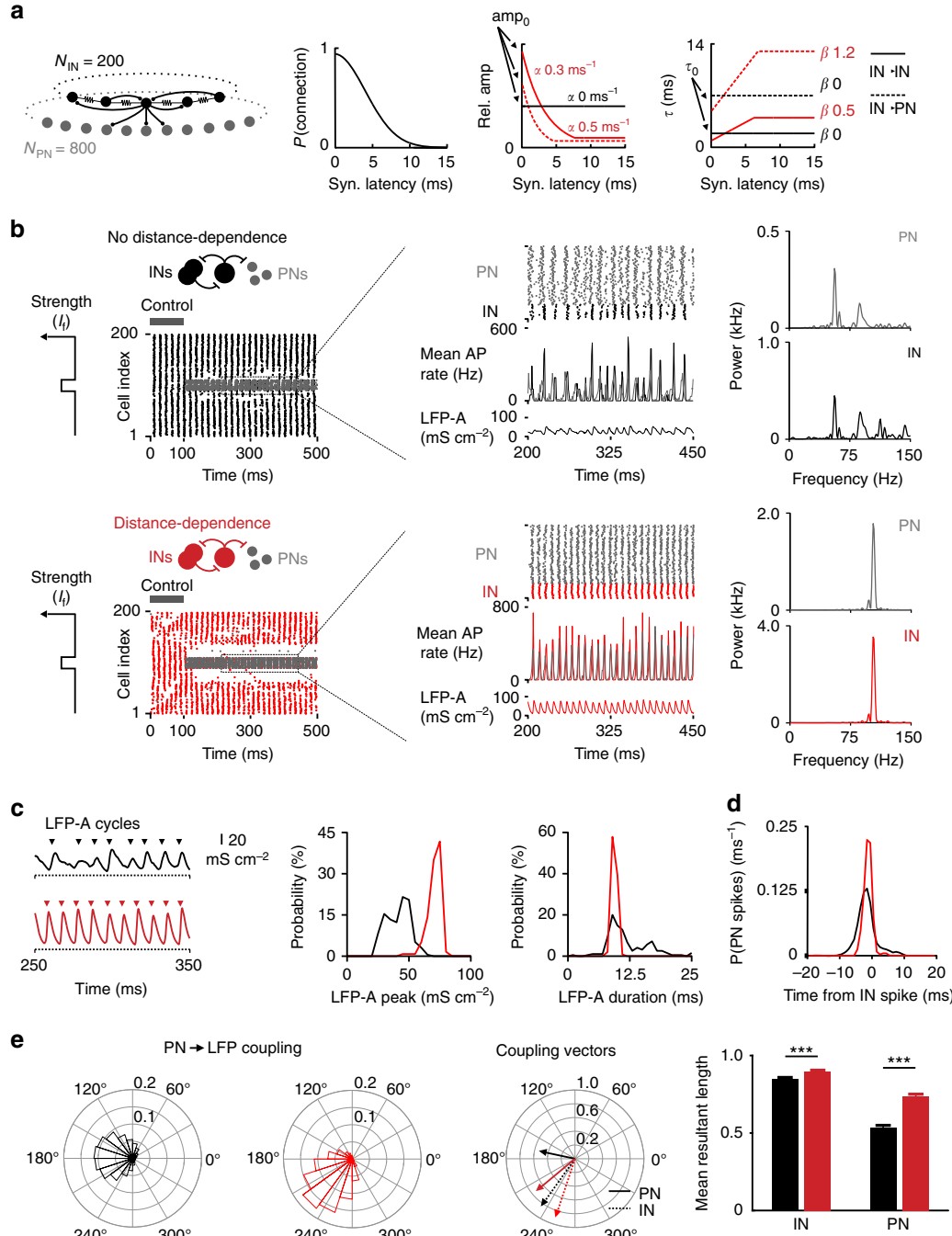

**Fig. 3** Distance-dependent inhibition supports focal gamma activity. **a** Standard properties of the IN–PN network model. Left, circuit diagram; Gaussian IN–IN connection probability (P). Middle, relative uIPSC amplitude distribution normalized to the closest-neighbor connection in a noDD (black) and a DD network model (red; dashed IN–IN, continuous IN–PN connections). Note, for the noDD model, IN–IN and IN–PN connections have the same normalized amplitude distribution. Right, distance-dependent distribution of $\tau$ in a noDD (black) and a DD network model (red). Values indicate default parameters for $\alpha$ and $\beta$. **b** Left, raster plots show gamma oscillations in both networks during the control period (PN spikes (gray) and IN spikes (black/red for noDD/DD) are overlayed; PN cell index is divided by 4 to account for the PN/IN ratio). For $100 \leq t < 500$ ms, focal excitation ($I_f$) was applied to the INs and PNs in the center. Middle, magnification of excited cells; PN and IN spikes are separated to improve visibility. Below, AP rate histograms (1 ms bins) and summed synaptic conductances as LFP-analog (LFP-A). Right, power density estimations from PN and IN discharges shown in the middle. **c** Left, representative LFP-As showing subsequent gamma cycles recorded from focally excited PNs in both models. Arrowheads indicate beginning of individual gamma cycles. Right, distributions of amplitude and length of individual LFP-A cycles from 10 simulation runs. **d** Probability (P) of PN spikes at different time points relative to the maximum of IN activity of the closest gamma cycle. Note, stronger PN–IN spike coupling in DD model ($N = 10$ simulation runs, spikes from 80 PNs). **e** PN coupling to the LFP-A. Left, mean normalized angle histograms of PN spikes to LFP-A phase (0° at LFP-A peak). Middle, mean resultant coupling vectors for IN (dashed) and PN (continuous) spikes, with angle and length corresponding to the phase and strength of single action potential coupling, respectively. Right, in the DD network PNs couple more strongly to the inhibitory LFP-A. Bars and error bars represent mean and SEM from 10 simulation runs (spikes from 80 PNs and 20 INs). ***$p < 0.001$; two-tailed, two-sample $t$-test

**Distance-dependent inhibition supports focal gamma bursts.** Which role does distance-dependent perisomatic inhibition play in the generation of focal gamma bursts? To address this question, we developed a neuronal network model consisting of 200 fast-spiking INs and 800 principal neurons (PNs). INs were interconnected by chemical inhibitory and electrical synapses and formed connections to PNs with experimentally driven properties and connectivity distributions[36–38] (Fig. 3a; see Supplementary Table 1). We compared two models: The DD network, containing the experimentally described distance-dependent inhibitory signaling at PII–PII and PII–GC synapses defined by DD coefficients $\alpha$ and $\beta$, and the non-distance-dependent (noDD) model with uniform distributions of amplitude and $\tau$ values of uIPSCs ($\alpha = 0$, $\beta = 0$; Fig. 3a). The strength of the average compound IPSCs and the mean $\tau$ were the same in both network models (balanced DD network model; "Methods"). Both networks were activated by Poisson-distributed excitatory synaptic events, which resulted in low-frequency gamma activity, driven by the IN population (Fig. 3b; control period; mean firing rates: $28.05 \pm 0.49$ Hz (IN, DD), $0.02 \pm 0.01$ Hz (PN, DD) versus $37.77 \pm 1.35$ Hz (IN, noDD), 0 Hz (PN, noDD)).

Lesions of the entorhinal cortex reduce gamma power and switch gamma frequency from the upper to the lower range indicating that a cortical excitatory drive is required for the emergence of high gamma activity in the dentate gyrus[7]. To reproduce high-frequency gamma bursts (Fig. 1d, e), we applied a focal excitatory drive ($I_f$) to a subset of 20 INs and 80 PNs, representing a cortical input (Fig. 3b). We observed rapid recruitment of a focal cell assembly which discharged at a significantly higher gamma frequency and power in the DD than the noDD model (power of firing rates: DD PN power 1.05 kHz at 106 Hz oscillatory frequency, DD IN 2.63 kHz at 106 Hz versus noDD PN 0.22 kHz at 57 Hz, noDD IN 0.45 kHz at 57 Hz; $p < 0.001$; mean firing rates: $24.30 \pm 0.25$ Hz (PN, DD), $97.18 \pm 1.00$ Hz (IN, DD) versus $14.47 \pm 0.71$ Hz (PN, noDD), $57.88 \pm 2.82$ Hz (IN, noDD); Fig. 3b, right). Moreover, gamma oscillations were restricted to the upper gamma frequency band in the DD but to a broader frequency range in the noDD assembly (frequency range [$f$] containing 80% of the total PN power in DD: $102 \le f \le 110$ Hz versus noDD: $54 \le f \le 116$ Hz; IN power in DD: $102 \le f \le 109$ Hz versus noDD: $56 \le f \le 127$ Hz; power spectra in Fig. 3b, right). Focal PN activity was more phasically modulated in the DD assembly as reflected by the sharp firing rate histograms (Fig. 3b, middle) and tight spike coupling between PNs and INs (Fig. 3d; standard deviation (SD) of time lags $1.3 \pm 0.1$ ms for DD assemblies versus $3.0 \pm 0.5$ ms for noDD PNs; $p < 0.001$). Improved PN–IN spike coupling could be explained by a more phasic inhibitory output during focal gamma oscillations. Indeed, the LFP analog (LFP-A), defined as the summed momentary synaptic conductances obtained from a group of adjacent cells, had a significantly larger amplitude and weaker fluctuation in its duration in DD than in noDD central cell assemblies (peaks DD: $70.6 \pm 5.4$ mS cm$^{-2}$ versus noDD: $41.2 \pm 9.2$ mS cm$^{-2}$, $p < 0.001$, Mann–Whitney $U$-test; cycle duration DD: $9.4 \pm 0.4$ ms versus noDD: $12.0 \pm 4.2$ ms, $p < 0.001$ for difference of variance, Ansari–Bradley test; Fig. 3c). In consequence, in the DD network, PN spikes were more strongly coupled to the inhibitory LFP-A (mean resultant coupling vector length DD: $0.737 \pm 0.013$ versus noDD: $0.536 \pm 0.013$, $p < 0.001$; Fig. 3e). These positive effects of distance-dependent inhibition on focal synchronization and PN spike timing could also be observed in an alternative, unbalanced version of the DD network, in which amplitude and kinetics of inhibitory synapses in the DD and noDD network were equal for closest neighbors (Supplementary Fig. 2).

Thus, distance-dependent inhibition enables focally excited microcircuits to effectively uncouple from the surrounding network and generate synchronous gamma bursts at high frequency and improved gamma power as well as tight control of PN spike timing. As focal gamma activity emerged from reciprocal inhibition between INs (interneuron gamma, "ING"), improved power of IN activity directly results from distance-dependent signaling at IN–IN synapses. However, to further evaluate the relative influence of DD at IN–IN versus IN–PN synapses on PN activity, we constructed chimeric networks with distance-dependent inhibition only in either of these synapse types (DD II and DD IE, respectively). In the resulting four network models (noDD, DD, DD II, and DD IE), we systematically varied the intensity of both the focal excitation and synaptic inhibition of PNs and analyzed focal network oscillations (Supplementary Fig. 3). PN firing rate and phase coupling were substantially altered after introducing distance-dependent signaling at both IN–IN and IN–PN synapses (Supplementary Fig. 3d, f). At lower PN firing rates due to weaker excitation and stronger inhibition, distance-dependent IN–PN synapses strongly improved PN coupling to the LFP-A. At higher PN activity levels caused by stronger excitation and weaker inhibition, distance-dependent inhibition at IN–IN synapses had a strong, indirect effect on PN spike timing by better synchronizing the IN network (Supplementary Fig. 3d–f). Thus, together, distance-dependent inhibition at IN–IN and IN–PN synapses facilitate synchronization of the IN activity and improve time-locked pacing of PN firing[26].

However, gamma activity can be generated by an alternative mechanism, which relies on excitatory feedback from local PNs to INs (principal neuron–interneuron gamma, "PING" model)[31, 39]. To assess the influence of distance-dependent inhibition on PING-mediated gamma oscillations primarily evoked by stimulation of PNs, we introduced PN–IN synapses and gradually increased their maximal conductance (Fig. 4). For increasing excitatory feedback strength, the focally excited cell assembly passed through three different dynamic regimes (Fig. 4a–h). For $g_{IE} = 0$ mS cm$^{-2}$, networks oscillated in the ING regime with higher frequencies >50 Hz and very high IN coupling to the LFP-A (Fig. 4a, g, h). On the other end, for $g_{IE} \ge 0.06$ mS cm$^{-2}$, network models were active in the PING regime with lower oscillation frequencies of ~40 Hz and weaker IN coupling to the LFP-A (Fig. 4d, g, h). In between, for $g_{IE} = 0.02$–$0.04$ mS cm$^{-2}$, the circuits seemed to operate in a transition phase with reduced synchrony and LFP-A coupling of neuronal firing (Fig. 4b, c). Interestingly, in the PING regime, DD networks showed a substantially stronger LFP-A power and better coupling of PN spikes to the LFP-A, while the IN coupling remained similar for DD and noDD circuits (Fig. 4g, h). These results indicate that, in contrast to the ING regime (Supplementary Fig. 3), only the DD in IN–PN connections enabled improved synchronization of PN firing (see also ref. [26]). We repeated these simulations in the chimeric DD II and DD IE networks and further confirmed that IN–PN but not IN–IN DD leads to higher spike timing precision of PNs in the PING regime (Fig. 4h). In summary, distance-dependent inhibition boosts the power of focal gamma activity and PN spike timing in both ING and PING regimes: during PING oscillations, only the DD in IN–PN synapses effectively improves focal synchrony. In contrast, in the ING regime DD in IN–IN synapses improves IN synchronization, which in turn together with the DD in IN–PN connections enables highly precise PN spiking (Fig. 4i, j).

**Improved focal gamma synchrony in DD circuits is robust.** Distance-dependence in IN–IN synapses plays a pivotal role during the emergence of highly synchronous focal ING but not PING oscillations. Due to the dominant inhibitory control of gamma activity and the extreme sparsity of GC firing in the

dentate gyrus[32, 34, 35], ING seems to be a relevant model for high frequency gamma activity in this brain region. Therefore, to examine how DD IN–IN synapses boost high frequency and power of focal gamma bursts, we concentrated on IN networks without excitatory PNs. This reduction of computational complexity further allowed us to disentangle the individual effects of distance-dependent amplitudes and $\tau$. First, we compared DD with noDD and shuffled IN circuits, in which synaptic parameters of the DD IN network were randomly permuted over inter-cell distances (Supplementary Fig. 4). Introducing a Gaussian shaped

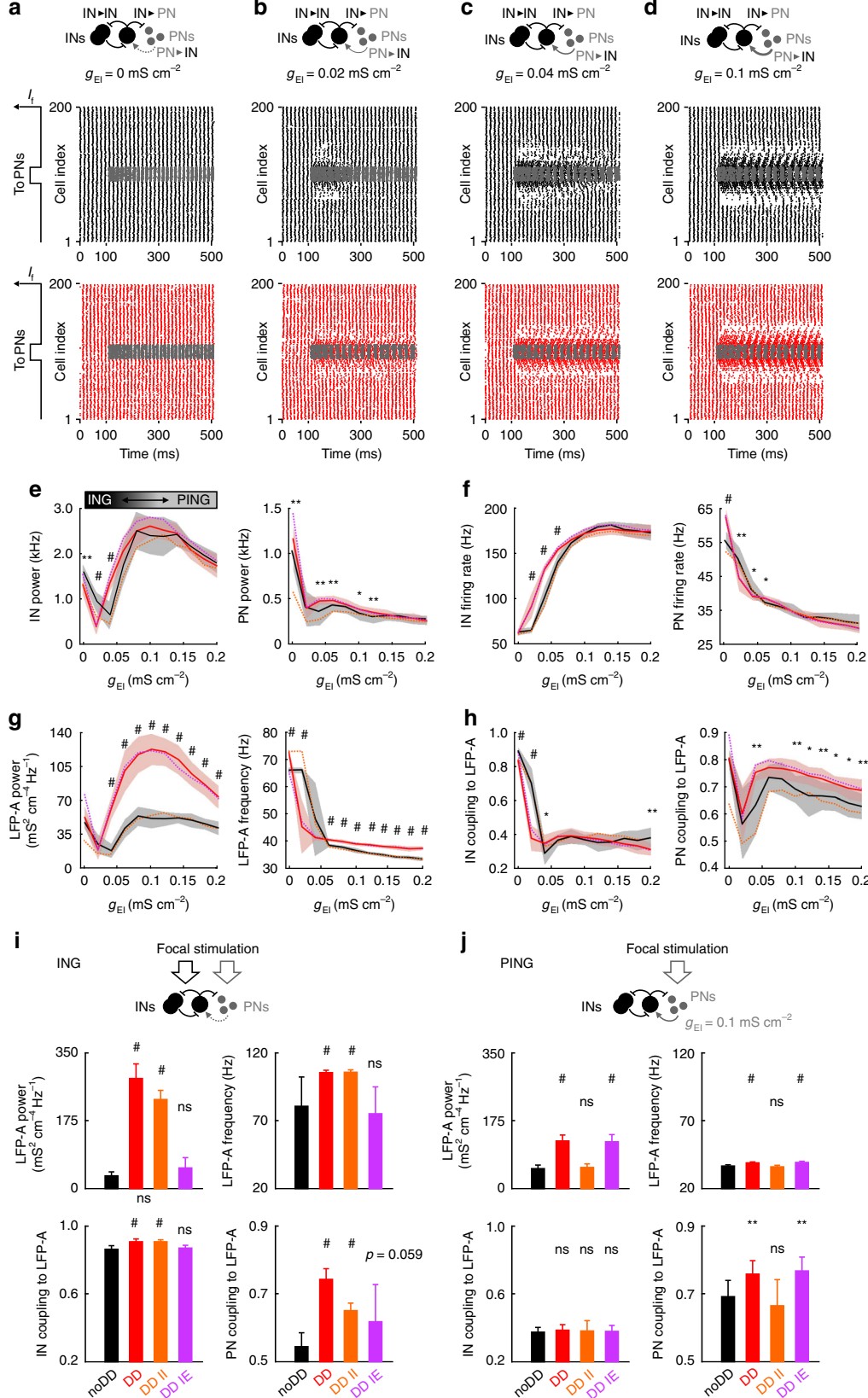

focal tonic excitation ($I_M$) to the network resulted in the emergence of focal gamma bursts at high gamma frequencies similar to our IN–PN model (Supplementary Fig. 4b versus Fig. 3b). We systematically altered various model parameters such as the strength of global and focal network excitation and synaptic inhibition (Supplementary Fig. 4d). For all conditions tested, gamma power was always higher in the DD than in the noDD or the shuffled network, indicating that boosting of focal gamma power was a robust finding, independent of the precise setting of the network parameters (Supplementary Fig. 4d, red continuous lines).

**Distance-dependent IPSC amplitude versus $\tau$ in focal gamma.** How does distance-dependent inhibition promote synchrony of focal gamma activity? To address this question, we first separated the two distance-dependent synaptic parameters, amplitude and decay time constant $\tau$, and examined their individual effect on the properties of focal gamma activity (Fig. 5). We quantified population coherence by a cross-correlation based measure for network synchrony ($\kappa$) and recruitment of cells by dividing each cell's firing rate by the local oscillation frequency ("Methods")[40],[41]. Starting from the noDD network (DD coefficients $\alpha = 0$, $\beta = 0$), we stepwise elevated the steepness in the DD of the amplitude amp ($\alpha$; DDamp network) or $\tau$ ($\beta$; DD$\tau$ network) (DD$^{\alpha|\beta}$; Fig. 5a). Changes in $\alpha$ or $\beta$ had qualitatively different effects on the properties of gamma oscillations. Increasing $\alpha$ markedly raised central coherence and central IN recruitment ($\alpha$: $0 \rightarrow 0.15 \rightarrow 0.3$ ms$^{-1}$; central coherence: $0.27 \rightarrow 0.36 \rightarrow 0.43$; central recruitment: $69.2\% \rightarrow 75.1\% \rightarrow 83.5\%$; Fig. 5a, left column) without affecting the periphery. In contrast, a rise in $\beta$ resulted in a significant reduction in network coherence and recruitment of cells in the periphery ($\beta$: $0 \rightarrow 0.125 \rightarrow 0.25$; peripheral coherence: $0.17 \rightarrow 0.12 \rightarrow 0.1$; peripheral recruitment: $44.3\% \rightarrow 37.2\% \rightarrow 34.6\%$; Fig. 5a, middle column) and mild changes in the center. The combination of both parameters boosted coherence of gamma activity and recruitment of INs in the center, reduced them in the periphery and thereby amplified the contrast between center and periphery (DD$^{0.3|0.25}$ versus DD$^{0|0}$: central coherence $= 0.32$ versus $0.27$, $p < 0.001$, Mann–Whitney $U$-test; central recruitment $= 74.8\%$ versus $69.2\%$, $p < 0.001$; peripheral coherence $= 0.14$ versus $0.17$, $p < 0.001$; peripheral recruitment $= 34.6\%$ versus $44.3\%$, $p < 0.001$; Fig. 5a, right column). Pairwise correlation measures are known to be sensitive to changes in the firing rate[42]. We, therefore, confirmed that our results are not caused by a direct effect of higher recruitment in DD networks on the measured coherence $\kappa$, and re-analyzed our data using the spike time tiling coefficient (STTC), an alternative method quantifying spike coherence, which is independent of neuronal firing rates[42]. We obtained comparable results using $\kappa$ or STTC (Supplementary Fig. 5, Fig. 5a).

Distance dependence in the properties of synaptic inhibition may vary in relation to the brain area and its developmental stage. We, therefore, broadly varied both DD coefficients and investigated their influence on the properties of network oscillations (Fig. 5b). Interestingly, for all variations tested, the center-to-periphery ratio of coherence and recruitment was always larger in the DD IN networks, indicating that this is a highly robust finding. While the high contrast in coherence was largely controlled by the DD in IPSC amplitude, increased recruitment contrast equally benefitted from DD in amplitude and time course.

**Inhibition is more periodic in gamma centers of DD circuits.** Previous studies showed that strength and duration of the total inhibitory conductance ($G_{inh}$) that an individual neuron receives by all of its convergent unitary inhibitory inputs ($g_{inh}$) are major factors determining the dynamics of network oscillations[40, 41, 43]. Thus, the individual effect of distance-dependent amplitudes and time courses of inhibitory signals on gamma coherence and interneuron recruitment in the center and the periphery of the DD network should be reflected in the properties of $G_{inh}$. To test this prediction we recorded $G_{inh}$ from individual central cells in the DDamp network and peripheral cells in the DD$\tau$ network during focally evoked gamma oscillations and compared the data with the noDD circuit (Fig. 6).

Inhibition in the center of the focal gamma activity showed a prominent oscillating pattern (Fig. 6a). However, the amplitude distribution of $G_{inh}$ of individual gamma cycles was shifted to significantly larger values in the DDamp network model ($p < 0.001$; Fig. 6b, top left). Furthermore, the duration of inhibitory cycles showed a markedly narrower SD ($p < 0.001$; Fig. 6b, bottom). Thus, the total inhibitory conductance is stronger and more periodic in DDamp than in noDD centers. Enhanced periodicity was further reflected in the increased autocorrelation amplitude ($p < 0.001$; Fig. 6b, bottom). To distinguish between the influence of strength and periodic shape of $G_{inh}$ on the observed improvement of central coherence, we reduced the effective unitary peak conductance (amp$_{effective}$) to 50%. Interestingly, coherence in the center was unchanged (Supplementary Fig. 6), indicating that enhanced periodicity of $G_{inh}$ is the primary mechanism underlying the improved synchrony of central gamma oscillations.

**Reduced synchrony of inhibition outside gamma foci.** Reduction in gamma synchrony in the periphery is mediated by the distance-dependent slowdown of IPSCs (Fig. 5a). To quantify this effect, we obtained $G_{inh}$ from cells in the periphery and characterized its properties during focal stimulation of the center (Fig. 6c, d). Peripheral inhibitory signals in the DD$\tau$ circuit

**Fig. 4** Distance-dependent inhibition boosts focal PING oscillations and PN spike timing. **a–d** Local rhythmic activity is induced by focal stimulation of a subgroup of only PNs in networks with different strengths ($g_{EI}$) of feedback excitatory synapses to INs (PN–IN). Top, schematic of the circuit. Bottom, raster plots showing the activity in the IN (noDD: black; DD: red) and PN (gray) population before and after stimulus onset at $t = 100$ ms for $100 \leq t < 500$ ms. **e–h** Central network activity for different $g_{EI}$ values. Black (noDD) and red (DD) continuous lines and shaded areas indicate means $\pm$ SD for 10 simulation runs. Dashed lines indicate means of 10 simulations in chimeric DD networks with distance dependence only in either IN–IN (DD II, orange) or IN–PN synapses (DD IE, purple). **e** Dominant oscillation frequency and corresponding power of mean momentary IN and PN firing rates. **f** Average firing rates of INs and PNs. **g** Dominant oscillation frequency and corresponding power of the recorded LFP-A. **h** Mean coupling strength of IN and PN spikes to the LFP-A. Note that for $g_{EI} = 0$ mS cm$^{-2}$, the network generates ING oscillations. With increasing feedback excitation, the network switches to PING, corresponding to a drop in the oscillation frequency (**g**, right; see also schematic in **e**, top left). Statistical analysis compares noDD and DD networks. **i** Average properties of focal oscillations in noDD, DD and the chimeric DD II and DD IE networks. To elicit ING oscillations, PN–IN feedback was switched off and both INs and PNs received focal stimulation (cf. Fig. 3b). Top, dominant oscillation frequency and corresponding power of LFP-A of the central network. Bottom, focal IN and PN coupling to the LFP-A. **j** Analysis of focal gamma activity resulting from PING-based synchronization, corresponding to **i**. Networks contained PN–IN feedback ($g_{EI} = 0.1$ mS cm$^{-2}$) and only PNs received the focal excitation. Bars and error bars indicate mean and SD of 10 simulation runs. Statistical analysis compares all three DD networks to the noDD circuit. *$p < 0.05$; **$p < 0.01$; #$p < 0.001$; two-tailed, two-sample $t$-test

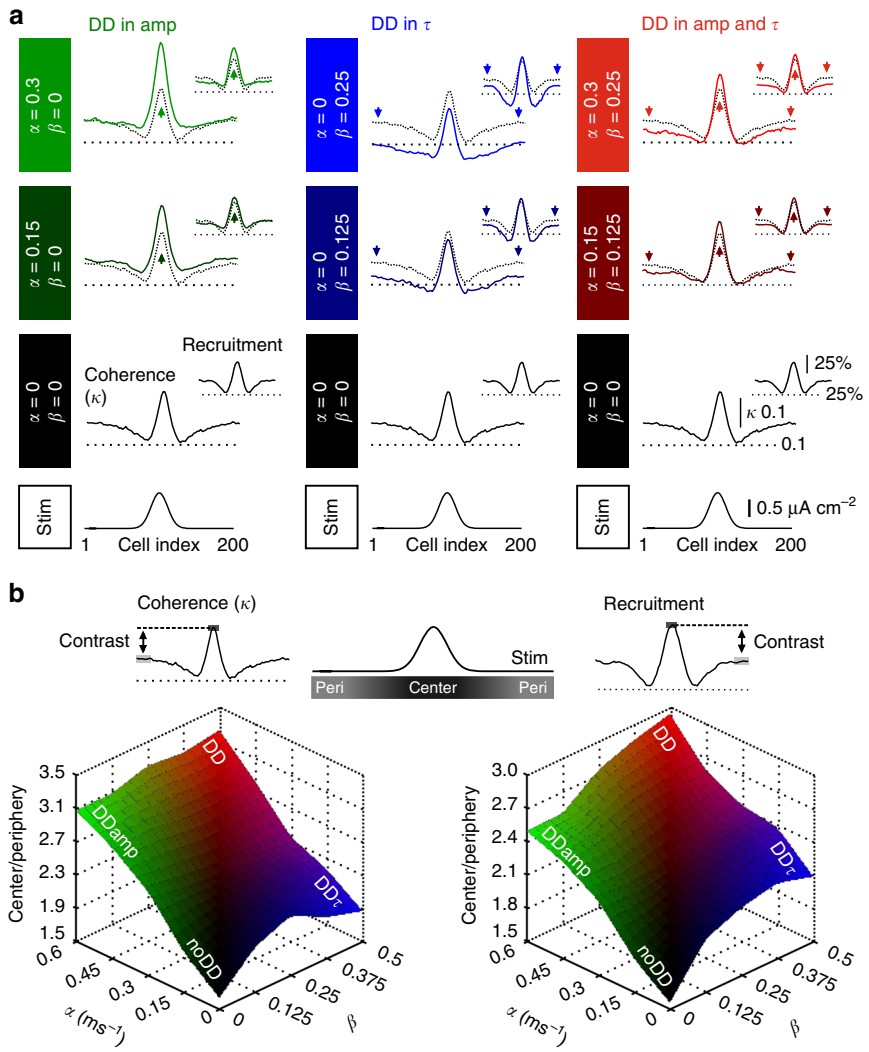

**Fig. 5** Distance dependence in amplitude and time course of IPSCs at IN–IN synapses synergistically increases the contrast between center and periphery in an IN network model. **a** Local analysis of the coherence $\kappa$ (left traces in each panel) and recruitment (smaller right traces in each panel) over 20 INs at each position in the network. Analysis was performed in the noDD network (black lines in each panel), in the DDamp (left column, green traces), DD$\tau$ (middle column, blue traces) and the full DD network (right column, red traces). For each column from bottom to top: Schematic illustrating the focal stimulation to the center of the network. Local $\kappa$ and recruitment in the noDD network. Comparison of local $\kappa$ and recruitment in the different DD networks (colored traces) with the noDD network (black dotted traces). Values of the respective DD coefficients $\alpha$ and $\beta$ increase progressively. Arrows indicate dominant effect of distance-dependent inhibition. **b** Top, schematic illustrates "center" (cells #91-110) and "periphery" (cells #1-30 and #171-200). Bottom, the center-to-periphery ratio of $\kappa$ (left) and recruitment (right) was plotted against DD coefficients. The noDD network ($\alpha = 0 | \beta = 0$) shows lowest contrast. Note, in combination, distance-dependent amp and $\tau$ result in the highest contrast between center and periphery (red). All data are averages of 50 simulation runs

showed major differences when compared to the noDD network. First, amplitude distributions of $G_{inh}$ during individual subsequent oscillatory cycles (Fig. 6c) were shifted to significantly smaller values ($p < 0.001$; Fig. 6d, top left). Second, the duration of inhibitory cycles showed a markedly larger variability ($p < 0.001$; reflected in a smaller autocorrelation amplitude $p < 0.001$; Fig. 6d, bottom), indicating that the total inhibitory signals in peripheral cells are less periodic in the DD$\tau$ network. Third, the conductance minimum at the trough (Fig. 6c, open triangles) was significantly larger, forming a "tonic form" of inhibition ($p < 0.001$; Fig. 6d, top right). Thus, distance-dependent time courses seem to impair the periodicity of inhibition and generate a tonic inhibitory component. Together, these effects directly reduce precision in spike timing and support silencing of INs (Supplementary Fig. 7).

**Multiple high-power gamma centers in DD networks**. Behavior-dependent spatial clustering of hippocampal principal cells in vivo has been proposed to underlie parallel network computation in the neocortex and CA1[10, 17, 19, 44], where several local gamma oscillations of similar frequency have been shown to coexist at different locations or at different frequencies with overlapping locations[10]. Our finding that distance-dependent networks promote the emergence of isolated gamma foci encouraged us to investigate network behavior during stimulation with multiple inputs in a network model containing both INs and principal cells (Fig. 7). We evoked a second active gamma center by an additional excitatory stimulus and systematically varied its location and magnitude (Fig. 7a). To measure the degree of interaction between both gamma foci, simultaneously recorded LFP-A signals were cross-correlated (Fig. 7a, bottom). Although the interaction between two adjacent gamma foci was comparably

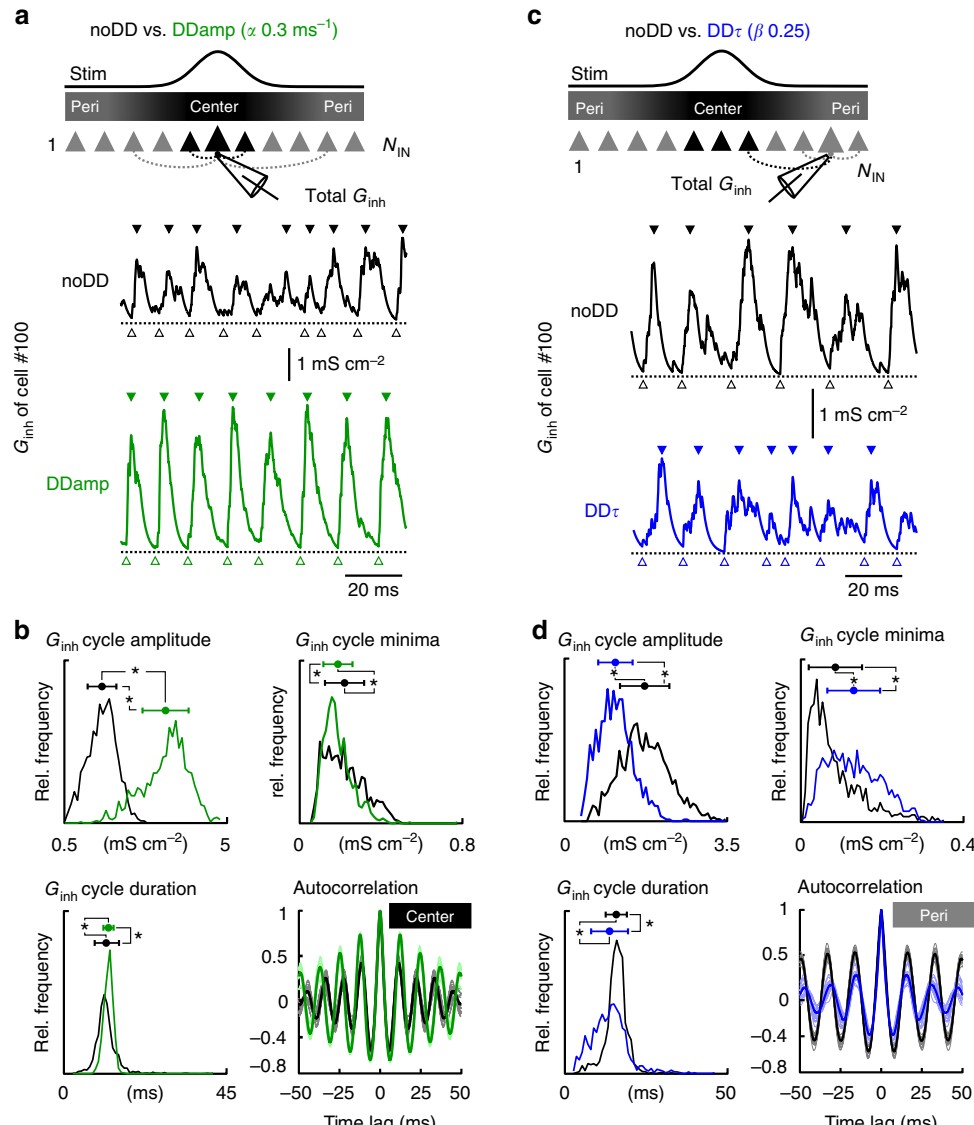

**Fig. 6** Distance-dependence in the amplitude boosts synchrony of central inhibition and distance-dependence in the time course desynchronizes inhibition in the periphery of an IN network model. **a** Representative total inhibitory conductance ($G_{inh}$) recorded from a cell (#100) exposed to focal stimulation (Stim) in the noDD (black trace) and the DD$^{0.3|0}$ (green trace) interneuron network model with distance-dependent amplitudes only (DDamp). Rhythmic patterns of $G_{inh}$ at gamma frequencies display peaks (filled triangles) followed by troughs (open triangles) in an alternating manner. **b** Distributions of amplitude, minima (open triangles in **a**) and duration of total $G_{inh}$ cycles. In the DD$^{0.3|0}$ network, mean amplitude of $G_{inh}$ is significantly larger ($p < 0.001$, Mann–Whitney U-test) and duration less variable (SD) than in the noDD network ($p < 0.001$, Ansari–Bradley test). Bottom right, the average autocorrelation of the $G_{inh}$ in central cells (#99-101) shows that periodicity of the $G_{inh}$ is higher in the DD$^{0.3|0}$ network. **c** In analogy to **a**, representative $G_{inh}$ recorded from a cell (#10) in the periphery of the noDD (black trace) and the DD$^{0|0.25}$ (blue trace) interneuron network model with distance-dependent time course only (DDτ). **d** Distributions of amplitude, minima and duration of $G_{inh}$ cycles. In the periphery, $G_{inh}$ in the DD$^{0|0.25}$ circuit displays smaller cycle amplitudes (top left), higher residual inhibition at the troughs (top right) and a broader standard deviation of $G_{inh}$ cycle durations (bottom left). The weaker periodicity of the inhibition in the periphery of the DD$^{0|0.25}$ network is further confirmed by the auto-correlogram (bottom right, blue). Points with error bars represent mean and SD. Data in **b**, **d** are based on five simulation runs

large in both networks (10-20 cell-to-cell distances; Fig. 7b), at >30 cell-to-cell distances cross-correlation between both centers rapidly declined in the DD network but remained on a similarly high level in the noDD circuit (Fig. 7b). This finding suggests a high capacity of the DD network for parallel information processing.

To test this prediction in biological networks we evoked local rhythmic activity patterns in the dentate gyrus of acute slices by puff application of 1.5 M KCH$_3$SO$_4$ to the molecular layer[28]. Highly synchronous gamma activities (30–50 Hz) were monitored with two LFP recording electrodes positioned in the gcl in

the vicinity of two puff pipettes (Fig. 7c and Supplementary Fig. 8). Locally evoked gamma oscillations lasted for up to 2 s, were highly focal and neither silenced nor boosted by distant stimulation (Supplementary Fig. 8a, b). They depended on inhibitory GABA$_A$ receptors but did not require fast excitatory AMPA receptor-mediated synpatic transmission (Supplementary Fig. 8b, c).The distance between both sets of stimulus- and recording pipettes was systematically changed (100–800 μm) and the cross-correlation of the two LFPs quantified (Fig. 7c, bottom). As in the simulated DD network but unlike the noDD model, maximal cross-correlation of both LFP signals declined with

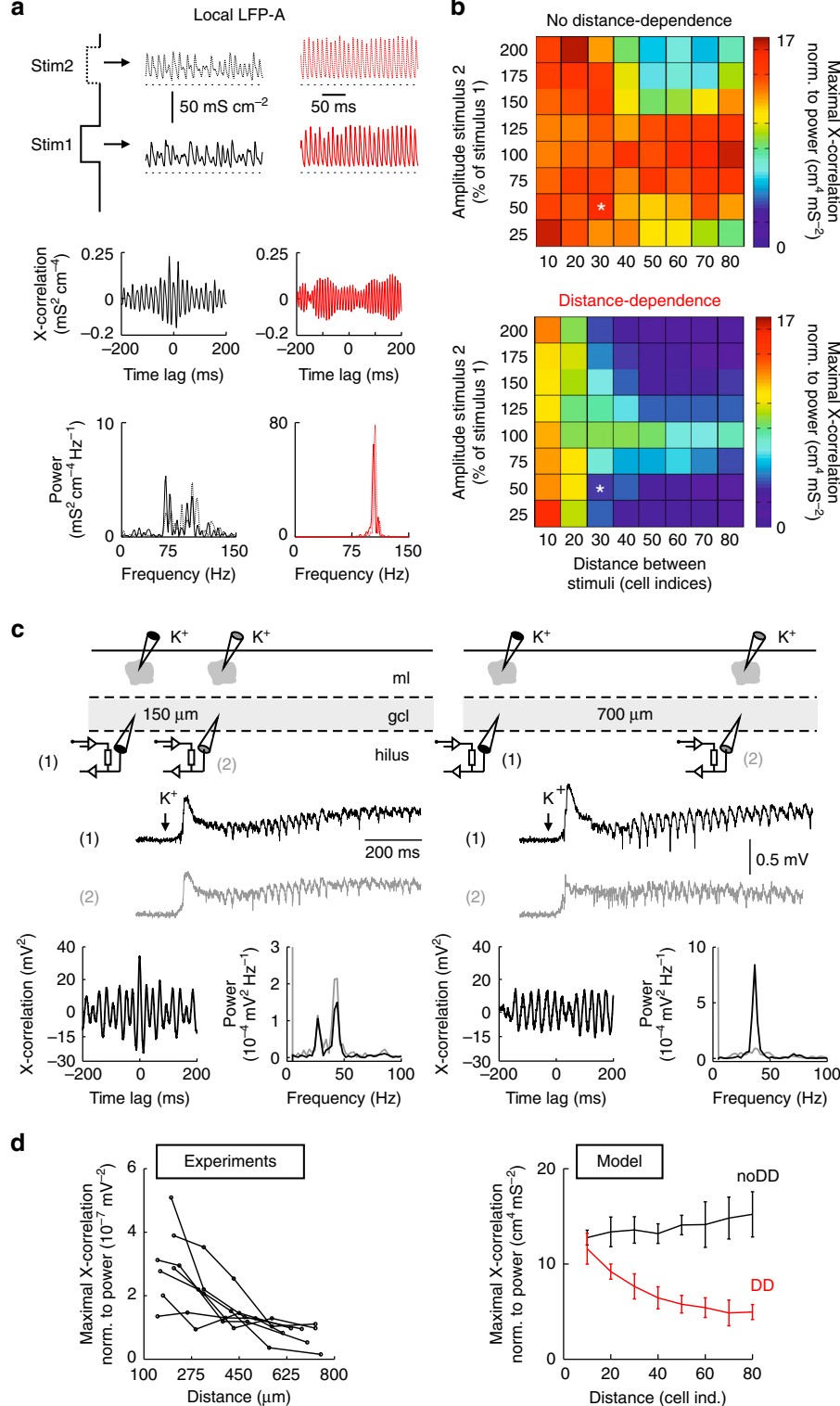

**Fig. 7** Parallel emergence of multiple local gamma oscillations in distance-dependent IN–PN network models and in vitro. **a** Top, LFP analog (LFP-A) recorded at the site of Stimulus 1 and 2 presented to the network at different locations (black: noDD, red: DD network). Middle, cross (X)-correlation of recorded LFP-A signals at stimulation site 1 and 2. Bottom, power spectra of the LFP-A recorded at stimulation site 1 (continuous line) and 2 (dashed line). The amplitude of the second stimulus was 50% of the first one; edge-to-edge distance between both stimuli was 30 cell indices. **b** Maximal LFP-A X-correlation was normalized to the 10–150 Hz power recorded at both stimulation sites (averages of 10 simulation runs). Asterisks indicate the situation depicted in **a**. **c** Top, schematic illustration of the experimental condition. Middle, representative LFPs recorded at distances of 150 and 700 μm and low-pass filtered at 1 kHz. Bottom, X-correlograms and power spectra are shown for the corresponding LFPs. **d** Left, summary plot representing normalized maximal X-correlation between both LFP signals as a function of the distance between the two recording electrodes. Right, comparison to the theoretical data from **b** (blue, noDD; red, DD network). Data points are averages of Stimulus 2 amplitudes 75–125% (10 runs each). Error bars indicate the SD

increasing distance between the two stimulation sites (Fig. 7d). Thus, our combined experimental and theoretical data suggest that distance-dependent inhibition supports the generation of independent gamma centers and, thereby, may increase the computational capacity of neuronal networks for parallel and independent information processing.

## Discussion

Here, we showed that functional heterogeneity in synaptic inhibition in networks of synaptically connected PIIs is not random but distance-dependent. The strength of synaptic inhibition declines exponentially whereas the decay time course increases linearly with distance between pre- and postsynaptic PIIs. This form of synaptic signaling is similar to the distance-dependent inhibition at PII–GC synapses in the dentate gyrus[26]. Thus, distance-dependent inhibition may define a general mechanism of PII-mediated perisomatic signaling.

Although the cellular and molecular mechanisms underlying distance-dependent inhibition at PII–PII synapses have not been fully explored in this study, our data indicate that they may be similar to the ones identified at PII–GC synapses[26]. The distance-dependent decline in strength at PII–PII synaptic signaling seems to be caused by a reduced number of synaptic contacts at remote target cells (Supplementary Fig. 1) as shown for PII–GC synapses[26]. This conclusion fits reconstructions of single intracellularly labeled basket cells in the dentate gyrus[45] and CA1[37], showing that output synapse density is high near the soma and decreases with distance. The distance-dependent slowdown of uIPSCs cannot be explained by differences in the somatodendritic location of synaptic contacts (Supplementary Fig. 1), but may be caused by a reduction in the expression of the GABA$_A$ receptor (GABA$_A$R) subunit α1 at distant release sites as observed at PII–GC synapses[26]. The α-subunit of the GABA$_A$R critically determines its deactivation and desensitization kinetics, with α1-containing GABA$_A$Rs displaying short opening times[46]. INs are known to express high levels of α1 postsynaptically in their GABAergic synapses, which are generally characterized by fast IPSC time courses[47, 48]. A reduction of the α1 content at distant output synapses and compensation by, for example, higher α2 levels could explain the DD of the IPSC time constants at PII–PII connections. Thus, we propose that anatomical and molecular changes with axonal distance are involved in distance-dependent PII-mediated inhibitory signaling.

Our multi-site LFP recordings in the DG of freely moving mice showed that gamma oscillations occur on different spatial scales, indicating that coherent neuronal activity can engage the whole network or only a spatially clustered subfraction of cells. This diversity in the spatial extent of gamma oscillations could only be discovered using high-resolution CSD analysis but not in traditional LFP recordings, which has important implications for future studies of the oscillatory structure of hippocampal but also other cortical areas. We found a clear relationship between the spatial size of gamma bursts and their oscillation frequencies, essentially converging to the dichotomy of global, low frequency (<50 Hz) versus local, higher frequency (>50 Hz) gamma activity. Although this specific relationship has to our knowledge not been shown for the hippocampus, the variability in spatial extent and frequency stands in good agreement with the existing literature on cortical gamma activity[6, 17, 18, 20, 30, 39]. For example, in the subiculum, two forms of gamma oscillations coexist, low (<50 Hz) and high (>100 Hz) gamma band activity[16]. While low gamma oscillations display robust spatial coherence and involve PN–IN–PN feedback loops[39], high-frequency gamma activity emerges focally and power declines rapidly with distance from the recording site[10, 13, 16]. This activity pattern is independent of

AMPA receptor-mediated excitation but requires fast GABA$_A$R-driven signaling[16]. The frequency of gamma activity depends on many factors, e.g., the strength of the excitatory drive to the network[17]. Additionally, the circuit mechanisms generating gamma oscillations strongly push oscillation frequency in either the low or higher frequency range[30, 39, 49] (Fig. 4): PING oscillations emerge through PN–IN–PN feedback loops leading to lower frequencies but also longer range synchronization[39, 50]. ING oscillations come about from mutual inhibition between INs and generally present with higher frequencies and lower robustness against heterogeneities in the excitatory drive[39, 49]. We, therefore, propose that the dichotomy of slow-global versus fast-focal gamma band activity can in part be explained by its mechanisms of emergence[20, 30, 39, 51] but also by the strength and the spatial structure of the excitatory input to the network[17, 43]. However, beyond this dichotomy, focal ING and focal PING oscillations have different minimal requirements regarding the structured excitatory input. For focal ING to emerge, afferent excitation to the IN network has to be spatially structured (Figs. 3 and 4i), while the drive on the PN population can be spatially uncorrelated. In contrast to focal ING, focal PING requires spatially correlated excitation of PNs, while INs inherit their spatial tuning from local feedback excitation provided by PNs (Fig. 4a–d, j).

Our computational analysis shows that distance-dependent inhibition at IN–IN and IN–PN synapses has different effects. Distance-dependent inhibition of PNs generates a more phasic inhibitory output and thereby improves spike timing of PNs during global and local, ING and PING activity (Fig. 4)[26]. In contrast, distance-dependent inhibition between PIIs is mostly effective during ING oscillations. It confers a unique spatial flexibility to the hippocampal network permitting the emergence of rhythmic activity on various spatial scales ranging from global to highly focal oscillations: it improves synchrony of focal gamma bursts, enhances their contrast to the activity in the surrounding network and thereby boosts the network capacity for parallel information processing in co-active cell assemblies. This spatial flexibility is mediated by a dual mechanism of distance-dependent inhibition. Firstly, a decline in synaptic strength improves the periodic shape of inhibitory signals in the center and reduces the interference of distant peripheral cells with the center, thereby supporting the isolation of central cells (Fig. 6a, b). Secondly, a slowdown of inhibition impairs periodicity and promotes a more tonic form of inhibition in the periphery, which desynchronizes the activity in the surrounding network (Fig. 6c, d). As a consequence, centers can oscillate in their rhythm without being pushed out of phase by inputs from the surround. In this way, afferent inputs generating focal synchrony are reliably gated to downstream networks and resist perturbations from the surround[52]. Moreover, improved contrast between focal gamma centers and their surrounding allows co-emergence of several hotspots (Fig. 7), thereby permitting parallel processing of information in the network[52]. Our in vitro experiments indicate that two gamma foci can remain independent at distances ≥500 μm (Fig. 7d). Based on anatomical estimates on the surface area of the dentate gcl along its total septotemporal extent (4.84 mm$^2$ in mice, 16.93 mm$^2$ in rats)[53] and assuming a homogeneous distance-dependent profile of perisomatic inhibition over the whole hippocampus, we propose that ≤20 independent dentate gamma foci can simultaneously exist per hemisphere in mice and ≤68 in rats.

Our data indicate that properties of synaptic communication can establish the basis for the emergence of spatiotemporally defined oscillatory patterns induced by a structured excitation. Indeed, in sensory cortices topographically organized afferent inputs lead to the generation of spatially defined gamma foci[17].

Similarly, associational and entorhinal inputs to the hippocampus and projections within the hippocampus have well-defined topographic distributions[54, 55]. Thus, differential activity of these inputs may give rise to functional maps with spatially clustered principal cell assemblies in the hippocampus[19, 44, 55–57]. Given the emergence of hotspots of gamma patterns in the subiculum[16], CA1[13], and neocortex[10, 17], we propose that distance-dependent inhibitory signaling may more generally support the power of focal gamma bursts in cortical networks and thus the parallel processing of information. Such a mechanisms may be of particular importance for pattern separation processes in the rodent dentate gyrus.

## Methods

**In vivo multi-site LFP recordings**. All in vivo recording experiments complied with the national and European legislation. Six- to 8-week-old C57/Bl6 mice of either sex were anesthetized with isoflurane (induction 3%, maintenance 1–2% in $O_2$ at an airflow of $1$–$2\,l\,min^{-1}$) and fixed in a stereotaxic frame (Kopf Instruments). Buprenorphine was injected subcutaneously during the surgery (0.05–0.1 $mg\,kg^{-1}$ body weight). Body temperature was kept stable with a heating pad (38 °C, Witte & Sutor GmbH). Craniotomies (1–2 mm diameter) were performed to implant 2/4-shank silicon probes (250/400 µm shank spacing, 16/8 recording electrodes/shank, 25/100 µm electrode pitch; Cambridge Neurotech/Atlas Neuroengineering) into the dorsal dentate gyrus (stereotaxic coordinates: 2 mm posterior, 0.75–0.8 mm lateral, 2.5–2.8 mm ventral of bregma; coordinates apply to the most medial shank). Reference and ground wires were connected to stainless steel screws (1 mm diameter) in the bone over the cerebellum. After surgery, animals were given 3–4 days to recover before recordings were performed.

For electrophysiological recordings, the animals were placed in an open field arena (26*42 cm) inside a sound-attenuating chamber (recording duration: 15 min). LFPs were recorded at a sampling rate of 30 kHz using a 32-channel amplifier system (RHD2000, Intan Technologies). After recording, the animals were deeply anesthetized with urethane ($2\,g\,kg^{-1}$). To identify recording locations, electrolytic lesions were made by briefly (~1 s) applying 10–50 V to each electrode. The animals were intracardially perfused with phosphate-buffered saline (~1 min) followed by 4% paraformaldehyde (~10 min). The brains were sectioned (slice thickness 100–200 µm) and stained with 4′,6′-diamidino-2-phenylindole. Only animals with confirmed recording locations in the dorsal dentate gyrus were accepted for analysis.

LFP data were analyzed with built-in and custom-made routines running in the Python 2.7 Spyder IDE. To obtain the $CSD_{n,t}$ at electrode $n$ and time point $t$, we employed an estimate based on recorded voltages at discrete electrode sites as

$$CSD_{n,t} = \frac{LFP_{n-1,t} - LFP_{n,t} + LFP_{n+1,t}}{\Delta z^2}, \qquad (1)$$

where $LFP_{n,t}$ is the voltage recorded on the silicon probe contact $n$ at time $t$, $LFP_{n-1,t}$ and $LFP_{n+1,t}$ are the voltages from neighboring sites at time point $t$, and $\Delta z$ is the inter-electrode spacing[58]. The obtained CSD signals of each mouse were bandpass filtered across narrow bins located in the gamma frequency range (30–150 Hz, 10 Hz bandwidth) using a 2nd-order Butterworth filter in forward and reverse direction to avoid phase shifts. The filter was generated with the scipy.signal.butter function applied to the data via scipy.signal.filtfilt. Bandpass filtered signals were converted to a z-score by subtracting the mean and dividing by the SD of the signal. Epochs of significant gamma activity in each frequency bin were identified as exceeding a threshold of two SDs above the mean of the root mean square of the signal (Fig. 1b). For each detected gamma period we computed the focality index

$$I_F = \left| \frac{Power_{electrode1} - Power_{electrode2}}{Power_{electrode1} + Power_{electrode2}} \right|, \qquad (2)$$

where $Power_{electrode1}$ and $Power_{electrode2}$ represent the root mean square signal of electrodes on two adjacent shanks inside the dentate gcl.

**Paired recordings between PIIs**. Data on the distance-dependent relationship between amplitude and τ with the spatiotemporal distance between connected PIIs stem from already published PII–PII paired recordings[36]. In brief, paired whole-cell patch-clamp recordings were performed in acute slices of the dentate gyrus of 20- to 25-days-old transgenic mice, expressing EGFP under the control of the PV promoter. Single action potentials were evoked in the presynaptic PII by short depolarizing current injections and postsynaptic uIPSCs were recorded in voltage-clamp with series resistance (≤10 MΩ) compensation enabled. Synaptic properties were determined from averages of ≥30 individual synaptic events. Recording temperature was 33–34 °C. For further details, please see ref. [36].

**Microscopical quantification of PII–PII synapses**. Single PIIs were intracellularly filled with 0.2% biocytin in acute transverse dentate gyrus slices of P18 Wistar rats. After filling, slices were fixed overnight in 4% paraformaldehyde at 4 °C and primary antibody (rabbit anti-PV, 1:1000, Swant) and secondary fluorescence stainings (PV: goat anti-rabbit Cy3, 1:1000; Jackson Immunoresearch; biocytin: AlexaFluor647-conjugate, 1:500; Invitrogen) were performed. Putative synaptic contacts between the biocytin-filled PII and remaining PV⁺ cells were identified as close appositions of presynaptic axonal boutons in close vicinity of PV⁺ somato-dendritic compartments by confocal microscopy (LSM710, Zeiss; Apochromat ×63 oil immersion objective). For all putative postsynaptic PV⁺ neurons in the slice, number of putative synaptic contacts, mean dendritic distance of these contacts from the soma and intersomatic distance to the biocytin-labeled cell were analyzed. Image analysis was done with ImageJ 1.51f. The experimenter was blinded during the analysis regarding the intersomatic distance between pre- and postsynaptic neurons.

**Network modeling**. The complete network models contained 200 fast-spiking INs and 800 regular spiking PNs arranged on a ring modeled in the NEURON 7.1 environment. Neurons were represented as single compartments. INs contained a leak conductance of $0.13\,mS\,cm^{-2}$ and voltage-gated Hodgkin–Huxley-type Na⁺ and K⁺ conductances with a resting potential of −65 mV. PNs contained passive, voltage-gated Na⁺, and three voltage-gated K⁺ (delayed rectifier, A-type, and M-type) conductances to imitate the regular spiking behavior of hippocampal principal cells. The mechanisms were adapted from Hemond et al.[59]. They rested at −75 mV.

INs were connected to four of their eight nearest neighbors via electrical synapses (transcellular conductance 0.01 nS)[36]. INs were spaced by 50 µm between adjacent cells. Every IN formed inhibitory connections with randomly chosen INs and PNs with a mean connectivity of 60 and 80 connections, respectively[29]. Connection probability of IN–IN and IN–PN synapses dropped with distance in a Gaussian manner (SD = 25 cell-to-cell distances). In a subset of simulations (Fig. 4), we implemented excitatory feedback from PNs to INs, with a mean connectivity of 50 presynaptic PNs per IN and a more narrow projection pattern (SD = 10 cell-to-cell distances). Synapses were modeled conductance-based with exponential rise and decay phases (Exp2Syn point processes) and with properties deferred from experimental data (this study, refs. [26, 36, 60, 61]). In the noDD model, unitary synaptic connections had the following properties in accordance to previous publications: peak amplitude = $0.2\,mS\,cm^{-2}$, rise time constant $\tau_{RT} = 0.16$ ms and decay time constant $\tau = 1.9$ ms (IN–IN); peak amplitude = $0.02\,mS\,cm^{-2}$, rise time constant $\tau_{RT} = 0.16$ ms and decay time constant $\tau = 7.0$ ms (IN–PN); peak amplitude = $0.05\,mS\,cm^{-2}$, rise time constant $\tau_{RT} = 0.2$ ms and decay time constant $\tau = 1.0$ ms (PN–IN). The reversal potentials were set to $E_{syn} = -55$ mV (IN–IN)[40], $E_{syn} = -65$ mV (IN–PN)[62], and $E_{syn} = 0$ mV (PN–IN). Synaptic delays consisted of a constant (0.5 ms) plus a variable component which scaled with connection distance along the circumference of the ring (conduction velocity $0.25\,m\,s^{-1}$)[26]. In the DD networks, peak amplitude and decay time constant changed with the distance between pre- and postsynaptic neuron according to the respective DD coefficients; noDD: $\alpha = 0\,ms^{-1}\,|\,\beta = 0$; DD: $\alpha = 0.3\,ms^{-1}\,|\,\beta = 0.5$ for IN–IN, $\alpha = 0.5\,ms^{-1}\,|\,\beta = 1.2$ for IN–PN (Fig. 3). These DD coefficients correspond to $\alpha$ and $\beta$ in Fig. 2b. Combinations of $\alpha$ and $\beta$ were varied over a broad range (Fig. 5). The default values for $\alpha$ and $\beta$ in network simulations were below the experimentally defined parameters to account for the conversion from a three-dimensional situation in the brain to the one-dimensional model. We set boundaries to the distance-dependent parameters amplitude and $\tau$ ($amp_\infty$ and $\tau_\infty$, $amp_0$ and $\tau_0$) to keep values in a realistic range. $Amp_\infty$ was set to ($0.1 \cdot amp_0$) to ensure quantal transmission at every formed synapse and $\tau_\infty$ was defined as 4 and 13 ms, the largest $\tau$ measured at PII–PII and PII–GC synapses, respectively (Fig. 2b, this study, and Fig. 1)[26]. The average strength ($amp_{effective}$) and time course of unitary inhibition critically influence network oscillations[36]. We designed two different versions of DD network models. In the balanced DD network, in a first step, $\tau_0$ was adjusted so that the average $\tau$ of all inhibitory synapses matched the noDD value of 1.9 and 7 ms for IN–IN and IN–PN connections, respectively. In a second step, $amp_0$ was chosen to set the average inhibitory strength (conductance integral) in DD networks equal to the noDD model with $amp = amp_{effective}$. Alternatively, in the unbalanced DD network, $\tau_0$ and $amp_0$ were set to the same values as in the noDD network, resulting in a smaller average inhibitory strength (Supplementary Fig. 2).

To control for the influence of heterogeneity on oscillatory dynamics, we constructed networks containing the same distribution of amplitudes and decay time constants as the DD networks, but shuffled in a random non-distance-dependent fashion (shuffled, Supplementary Fig. 4).

In each simulation run, neurons were launched with membrane resting potentials randomly picked from a uniform distribution on the intervals −63 to −67 mV for INs and −73 to −77 mV for PNs. Next, cells were activated in a time window of 50 ms with uncorrelated trains of Poisson-distributed excitatory synaptic events (background drive) with randomized onset times and heterogeneous peak amplitudes ($\tau_{rise} = 0.1$ ms, $\tau_{decay} = 2$ ms, amp = $0.01\,mS\,cm^{-2}$, CV of amp = 0.2 for INs and PNs; default frequencies: 300 Hz for INs and 500 Hz for PNs). 100 ms later, synapses were switched on and the network was left unperturbed for >350 ms. The last 100 or 200 ms of this initial period were used as

a control period without focal stimulation. The beginning of this control period was defined as time point $t = 0$ ms. After the control period, a focal excitation was applied to the network: In the complete IN–PN network, squared pulses of Poisson trains of excitatory conductance changes (CV of amp = 0 for INs and PNs) were targeted to 20 INs and 80 PNs (Fig. 3b); in the reduced IN network, Gaussian shaped focal tonic excitation was applied to the circuit with an amplitude $I_M = 1$ μA cm$^{-2}$ at the center and SD of 10 cell-to-cell distances (Fig. 5a, b).

**Quantitative analysis of oscillations**. For the quantification of oscillatory activity, different parameters were calculated on the basis of a spike matrix that indicated which cell has fired an action potential at which time bin with 1 ms temporal resolution. Coherence ($\kappa$) was determined by a pair-wise correlation measure and averaged for all cells under investigation[40]. Mean firing rate histograms were obtained from the spike matrices to determine power, recruitment, and frequency of network oscillations. Mean firing rate histograms were subjected to power spectral analysis using MATLAB's periodogram algorithm and the maximal oscillation power and the dominant frequency were identified as the coordinates of the highest peak in the resulting power spectrum estimate. The unit of firing rate power was given as kHz (i.e., kHz$^2$ kHz$^{-1}$). Recruitment was calculated by dividing the frequency of the ongoing oscillation in the area of interest by the actual firing frequency of individual cells. For the complete network, the total momentary synaptic conductance in the network was summed up resulting in a local field potential analog (LFP-A). In a subset of analyses, LFP-A was subjected to power density estimation as described above and cross-correlation analysis using MATLAB's xcorr function. The STTC[42] as an alternative firing rate-independent coherence measure was determined in a spike matrix with a temporal bin size of 0.2 ms. For any two cells A and B in the network, a spike in cell B would be considered synchronous to a spike in cell A, if it occurred in a time window of $t$(spike A) $\pm \Delta t$. Thus, $\Delta t$ represents a free variable directly affecting STTC. We therefore systematically varied $\Delta t$ to show the robustness of the results (Supplementary Fig. 5b). For the data in Supplementary Fig. 5a, $\Delta t$ was set to 1 ms.

We analyzed oscillations on a local network level. The spike matrix was obtained from the time frame of stimulation (200 ms < $t$ ≤ 450 or 500 ms). In order to calculate local synchrony parameters, for every value, 20 adjacent INs (and 80 PNs in the case of the complete IN–PN network) were grouped and their spike matrix was subjected to the aforementioned analysis procedures. This spatial analysis window was slid over the whole network in steps of 2 cell-to-cell distances. Principle results were not affected by different window sizes. All mean power, $\kappa$, and $f$ values are averages of 10 runs for the complete IN–PN network and 50 runs for the reduced IN network.

At different points in the course of the paper, we refer to a "central" versus a "peripheral" region of the network in relation to the focal stimulus. The borders of these regions cannot be explicitly drawn and they also depend on the DD coefficients. We, therefore, define the 20 cells receiving strongest stimulation (#91 to #110) as "central" and the cells #1 to #30 and #171 to #200 as "peripheral", if not stated otherwise.

**Spike to LFP-A coupling analysis**. LFP-A traces were bandpass filtered in the broad gamma range (30–150 Hz) and instantaneous phases were calculated as the arctangent of the ratio of imaginary and real part of the Hilbert transformed filtered LFP-A signal[63]. For every spike of the stimulated IN and PN cell group, the corresponding instantaneous phase was evaluated. Mean coupling phase and strength were analyzed using the CircStat toolbox[64] for MATLAB. The mean resultant length of the coupling vector was interpreted as spike to LFP-A coupling strength.

**Power and periodicity analysis of total inhibitory conductances ($G_{inh}$)**. To quantify the power of $G_{inh}$ we recorded the inhibitory signal in defined probe cells. In Fig. 6 and Supplementary Figs. 6 and 7, the probe cells representing the central region were INs #99, #100, and #101, whereas the peripheral region was represented by cells #1, #5, and #10. Their respective mean values are averages from three cells of the respective region over five randomized runs. $G_{inh}$ was recorded over 1 s long stimulus periods, normalized to their mean values and subjected to power spectral density analysis. The normalized power was used as a measure of the periodicity of the inhibitory signal independent of its average size. Auto-correlations of $G_{inh}$ (Fig. 6b, d) were analyzed using MATLAB's xcorr function.

**In vitro gamma oscillations**. Transverse hippocampal slices (400 μm) were cut from brains of 17- to 23-days-old Wistar rats of either sex with a VT 1200S (Leica Microsystems) vibratome. Animals were decapitated in accordance with national legislation and European guidelines. Acute hippocampal slices were superfused in an artificial cerebrospinal fluid (ACSF) consisting of (in mM) NaCl 125, NaHCO$_3$ 25, KCl 2.5, NaH$_2$PO$_4$ 1.25, D-glucose 25, CaCl$_2$ 2 and MgCl$_2$ 1 (equilibrated with 95%O$_2$/5%CO$_2$) for 30 min (34 °C) and then stored at room temperature (21–22 °C). Recording pipettes (wall thickness: 0.5 mm; inner diameter: 1 mm) were pulled from borosilicate glass tubing (Hilgenberg; Flaming-Brown P-97 puller, Sutter Instrument) and filled with ACSF for extracellular field potential recordings using an EPC 10 quadro (HEKA Elektronik Dr Schulze GmbH, Germany) or Multi-Clamp 200B amplifier (Molecular Devices). One or two puff pipettes (tip diameter

~2–4 μm) filled with 1.5 M KCH$_3$SO$_4$ were positioned in the ml of the dentate gyrus at different positions along the transverse axis. Furthermore, one or two extracellular electrodes (pipette resistance ~1–2 MΩ) recorded the LFP in the gcl next to the puff sites. Local depolarization was induced by pressure ejection of K$^+$ solution (12–20 psi for 40–100 ms). In vitro oscillation experiments were performed at room temperature. For correlation analysis, LFPs were cross-correlated and maximal cross-correlation was determined. To normalize the results to the strength of the recorded oscillations, we obtained power spectra using MATLAB's pwelch function and extracted the total 10–100 Hz power for both recording sites. Power spectra of control traces before stimulation were subtracted from power spectra obtained from recordings during stimulation to remove 50 Hz artifacts. Maximal cross-correlation was finally divided by the product of both total power values. For pharmacological experiments, the amplitude of low gamma frequency-filtered traces (20–60 Hz) before and after puff application was obtained using Hilbert transformation (scipy.signal.hilbert). After the acquisition of baseline gamma activity upon puff stimulation, NBQX (20 μM) or zolpidem (1 μM) were bath-applied for at least 6 min. Gamma oscillation frequency was determined as the peak in power spectral density distributions obtained using Python's matplotlib.mlab.psd function.

**Statistics**. Experimental values are given as mean ± standard error of the mean and data from modeling experiments as mean ± SD, if not stated otherwise. Statistical analysis was performed using either SigmaPlot 11 (Systat Software Inc., Chicago, IL, USA), MATLAB 7 (The Mathworks Inc., Natick, MA, USA) or Python's scipy.stats package. All further data analysis was done in MATLAB (except for in vivo LFP recordings and in vitro pharmacology, see above). For correlation analysis, Spearman's correlation coefficient $\rho$ was used. If not indicated otherwise, the difference in the mean of two samples of data was tested by a two-tailed unpaired $t$-test if the samples were normally distributed. Normality was tested using Lilliefors or Shapiro–Wilk test. If the normality test failed, the non-parametric Mann–Whitney $U$-test (independent samples) or the Wilcoxon signed rank test (paired comparisons) was employed. At defined places, the difference in the variance of data samples was tested using the Ansari–Bradley test. For the remaining group comparisons, sample variances were in a similar range. Sample sizes were adjusted accordingly to the expected effect size so that the gain in statistical power was traded off against the specific costs. This was particularly relevant for the in vivo recording experiments where the sample size of four mice was reasoned to be both necessary and sufficient to obtain reliable results. All experimental results were included in the paper.

**Data availability**. All data including codes for neuronal network models and analysis scripts are available from the authors. The NEURON code building the standard neuronal network model can be obtained from the ModelDB database[65] at http://modeldb.yale.edu/229750 (accession number: 229750). To obtain further experimental or theoretical data sets or analysis scripts employed in this study, please contact the authors directly.

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

## Acknowledgements

We thank Drs Imre Vida and Pepe Alcami for critically reading initial versions of the manuscript and Drs Ad Aertsen and Josef Bischofberger for fruitful discussions at earlier stages of the project. We thank Dr Patrick Ruther and Tobias Holzhammer for contributing silicon probes. This study was supported by grants from the Lichtenberg Award (Volkswagenstiftung; M.B.), the Schram Foundation (M.B.), the German Research Foundation (DFG BA 1582/2-1 and FOR2143, M.B.), the Excellence Initiative, GSC-4 Spemann Graduate School (M.S.), Brain-Links Brain-Tools, and the Cluster of Excellence EXC 1086 (M.B.). This project has received funding from the Fond zur Förderung der

Wissenschaftlichen Forschung (P 24909-B24) and the European Research Council (ERC) under the European Union's 7th framework program with grant agreement number 268548 (P.J.).

## Author contributions

M.S., J.-F.S., P.J., and M.B. contributed to the conceptual ideas of the study. M.S. and M.B. designed in silico and in vitro experiments. M.S., J.-F.S., and M.B. designed in vivo experiments. M.S. conducted and analyzed computational simulations. M.S. and J.-F.S. conducted and analyzed in vitro oscillation experiments. J.-F.S. performed in vivo experiments and analyzed the data. M.S., J.-F.S., P.J., and M.B. discussed the obtained data and wrote the manuscript.

## Additional information

**Competing interests:** The authors declare no competing financial interests.

