## [Peer Review File · Nature Communications]

Reviewers' comments:

Reviewer #1 (Remarks to the Author):

This paper provides a computational exploration of the possible adaptive significance of the finding that unitary IPSCs among fast-spiking interneurons in the dentate gyrus (defined as “perisomatic-innervating”) show both a decrease in peak amplitude and a decrease in decay rate with the distance separating the cells (with the IPSC onset latency used as a surrogate measure of distance). They show by modulating separately the amplitude-distance relationship and the tau-distance relationship that the two have complementary effects on the ability of a simulated network to generate robust local and independent oscillatory assemblies when presented with two spatially-separated ‘humps’ of excitation. This is interesting and conceptually novel. The authors then attempt to relate the findings to in vitro evoked gamma bursts and show that focal oscillations are more likely to synchronize when they are close together. This is not surprising, and the phenomenon could be achieved simply by restricting the connectivity of a network of inhibitory units in a distance-dependent manner (after all, transecting the tissue would achieve the same phenomenon). Finally, they show that brief bursts of gamma can occur in freely moving animals. Again, this is relatively remote from the simulations. The authors speculate about the number of independent gamma foci that can occur on the basis of the distance-dependent amplitude and tau relationship.

1. The figures are not easy to read because they are extremely small (although carefully prepared, and the manuscript is well written).

2. The most obvious limitation of the study is that the distance-dependence of connectivity is difficult to infer from brain slices because many axons are cut. The authors ignore this variable and simply assume a Gaussian distribution in the probability of interneurons connecting in their model. As pointed out above, the ability of a network to generate local gamma, and uncouple foci from one another, could be modulated by varying the connectivity. The local connectivity of interneurons in the model is however at variance with the vast majority of the literature on simulated oscillations, where interneurons are assumed to receive and send connections more broadly than principal cells, and therefore are poorly tuned to stimulus.

3. The central proposal that the spatial functional connectivity of PIIs supports local oscillations for information processing implies that physically neighboring neurons are more likely to participate together in representing information. Although this is the case in motor and sensory cortices, I am not aware of any evidence to support this in the hippocampal formation.

4. In Figure 1 the authors compare the DD situation to the no-DD situation, but this is applied both to PII-PII signaling and to PII-principal cell signaling. They do not separate out the roles of DD versus no-DD for PII-PII specifically. This would require exploring all 4 permutations of DD versus no-DD for PII-PII and PII-principal cells.

5. Page 5: the gamma frequency is given in some places as kHz. I presume this is the summed firing of many simulated neurons, but what is the average firing rate of each cell?

6. Page 5: “significantly weaker fluctuation in its duration...” The supporting statistics do not seem to address the difference in fluctuation, or else they are ambiguous.

Reviewer #2 (Remarks to the Author):

The study by Struber et al. aims to understand the role of distance-dependent inhibition in the generation of focused gamma oscillations in the dentate gyrus using in vitro and in vivo

experiments, complemented by computational modeling.

This is an elegant and important study that will likely have significant impact on the field. The results are convincing and novel. The paper is well written and the interpretation of the data is solid. I don't have major concerns; my comments below raise relatively minor points that the authors should be able to address.

1) The authors measure the amplitude and kinetics of PII-PII inhibitory synaptic currents in somatic voltage clamp. As the authors are well aware, cable properties and poor voltage control of dendrites attenuate the amplitude and slow the decay of synaptic currents generated in dendrites and measured at the soma. Therefore, the observed distance-dependence of IPSC amplitude and decay kinetics could be a consequence of a gradient of synapse locations. I.e., if the axons of PII neurons make local synapses preferentially near the soma of proximal PII neurons, and distal synapses preferentially on the dendrites of distal PII neurons, IPSCs recorded at the soma will appear smaller and slower for the dendritic inhibitory events. Do the authors have histological data to address this possible mechanism? If synaptically connected PII interneurons were filled to label dendrite and axon, it would be sufficient to quantify co-localization of axon and dendrite for proximal and distal pairs. In many ways, this is a trivial point, but it should be addressed.

2) Nitz & McNaughton, 2004 seems like not a sufficient justification for discarding PN-IN networks in favor of IN-IN networks. That study showed increased firing rates of interneurons during novelty. The paradigm and mechanisms explored in that study are orthogonal to focal gamma generation. The effects on gamma frequency and power of DD vs. noDD in the model are very weak under conditions of physiological levels of feedback excitation (Supp. Fig. 1). I would rather have seen the sensitivity analysis in Fig. 2 for IN-PN networks.

3) In the discussion it is stated that high frequency gamma "is independent of AMPA receptor-mediated excitation but requires fast GABAAR-driven signaling." The evidence for this seems rather weak, based on bath application of DNQX (Jackson et al., J. Neurosci., 2011). DNQX and CNQX are partial agonists of neuronal AMPARs that directly depolarize interneurons (Maccaferri & Dingledine, Neuropharmacology, 2002; Menuz et al., Science, 2007). If this is an important point for the study, the authors could test if NBQX, a true competitive antagonist of AMPARs, blocks gamma in their slice model.

4) In the previous study (Struber et al., 2015), the $\alpha 1$ -subunit-selective GABAAR agonist zolpidem was used to normalize the decay kinetics of proximal and distal IPSCs. What does zolpidem do to gamma frequency and power in the slice model?

5) I appreciate that it is no trivial endeavor to combine in vivo, in vitro, and in silico experiments. However, the flow of the current manuscript is a bit disjointed. Ideally, the paper would start with the observation of focal gamma in vivo, exploit the slice model of focal gamma to tease apart mechanisms, and use the model to make experimentally testable predictions that could be confirmed either by new analysis or perturbation of the experimental system.

Reviewer #3 (Remarks to the Author):

In vivo recordings from primate visual cortex have established that the frequency and spatial coherence of gamma-frequency oscillations are stimulus-dependent. The functional significance of such changes in gamma synchronization continues to be debated, but it is clear that our fundamental understanding of how cortical circuits operate must include an explanation of such activity-dependent changes in dynamic coupling.

In this manuscript, Struber et al show that local and global patterns of gamma synchronization also occur in the rodent dentate gyrus in vivo. Using paired recordings from perisomatic-targeting interneurons and extensive computational modelling, they suggest that distance-dependent properties of GABAergic synapses enable/enhance the activity-dependent spatial divergence of gamma synchronization.

It is well-established that conduction delays, inhibitory conductance and synaptic decay constants influence the spatiotemporal patterns of fast network oscillations (eg Brunel & Wang, 2003; PMID:12611969; plus the authors previous work). In a recent modelling study of the visual cortex, it has also been suggested that the frequency and spatial spread of gamma oscillations depends on both the difference in drive and coupling strength between local circuits (Lowet et al, 2015; PMID:25679780). However, the role of distance-dependent of GABAergic signalling may be an important new concept, particularly for explaining focal gamma oscillations in inhibitory networks.

Main concerns:

1) The authors show that the peak amplitudes and decay time constants of uIPSC vary with axonal distance (Fig 1b), which is contrary to the assumption that distal synapses would be less likely to occur, but have similar properties. To test the effects of 'distance-dependent inhibition' (DD) vs noDD in network models, the conductance integral is kept constant. This means that the amplitude is always larger and/or decay time constant always faster for local connections in the DD vs noDD model. In the case of Fig 1d, this means for the DD network there is strong fast inhibition at the centre of stimulation, and slow lateral inhibition (completing shutting down neighbouring neurons; removes interference from conduction delays and variance in synaptic decay constants), which has understandable effects on frequency and synchronization (eg Brunel & Wang, 2003).

It is reasonable to examine the effects under conserved average inhibitory synaptic strength, and the authors may feel that they have actually demonstrated whether this interpretation is correct or not. However, I think there should also be comparisons with DD and noDD networks that have the same absolute local amplitudes/decay time constants, which I believe was the null hypothesis for the uIPSC data. This will clearly have effects on baseline and/or stimulus-induced activity, but can be compensated for by changes in tonic GABA conductance / excitatory drive (rather than parameters of synaptic inhibition), and could establish the importance local vs peripheral parameters in the balanced model presented.

2) Fig 4 - The noDD model suggests that distance does not have a major effect on gamma cross-correlations for 2 equal stimuli separated by up to 800 micron. This fits with the connectivity probability shown in Fig 1c (along with x axis of Fig 1b; $\rightarrow \sim 100\%$) – how was this determined for the model?

Whatever the starting conditions, how changes in local and peripheral inhibition affect the localization of gamma activity seems readily interpretable. However, I do not understand the experimental manipulations / analysis:

i) What is the spatial resolution of the K+ pressure ejection, and what is the activity induced at each site alone

ii) What is the spatial resolution of the LFP recording (, and what is the activity induced at each site alone)

iii) At 700 microns is LFP2 suppressed by site 1 stimulation? – this is the example recording, but the LFP recordings look very different in amplitude, which is not analyzed. What was the response to single site stimulation? How does this relate to the Stimulus 2/ Stimulus 1 ratio in Fig 4a?

iv) I assume the cross-correlation coefficient is taken at lag 0. This must be higher at closer spaced LFP electrodes. However, the side bands look more prominent for wider spaced stim/LFP electrodes, which would be more indicative of rhythmic synchronization.

Other points:

1) The firing rates in the models of interneurons and principal cells are very high, with each neuron

likely showing strong gamma side peaks in the spike time autocorrelations even under baseline conditions.

Page 4, line 2 – ‘principal cell activity is extremely sparse’ – this is not reflected in the model (Fig 1). The activity of the interneurons is not representative of in vivo spike patterns either. Is it possible for DD inhibition to be effective with sparse firing?

It should be noted that the method of analyzing coherence based on pairwise comparisons depends on spike rate (are they active every cycle of a network rhythm), as well as synchrony. This is covered by the additional analyses of LFP-A and spike coupling in Fig 1, but these additional analyses are not always presented for parameter variation, and may be more important measures with low spike rates.

2) Figure 2a, left panel– plots of cell index and coherence do not appear to be vertically aligned.

3) The details of the electrophysiology experiments do not appear to be provided.

4) Supplementary Fig 1 – I do not see consistent effects of the DD vs noDD network with E-I connections, and it is not clear this has been analysed statistically. I would assume that E-I connections should be able to break down any effects of DD inhibition, at least to some extent, but this not preclude it from being an important factor in the operation of inhibitory networks.

Point-by-point response to the reviewer's comments:

We thank the editor and the reviewers for praising the novelty and importance of our work ('This is interesting and conceptually novel', '...an elegant and important study...', '...an important new concept...') and the quality of the manuscript ('...paper is well written and the interpretation of the data is solid..'). The reviewers had some questions and formulated proposals for changes of the manuscript as well as additional data analysis to which we reply point-by-point:

Reviewer 1 (Remarks to the Author):

This paper provides a computational exploration of the possible adaptive significance of the finding that unitary IPSCs among fast-spiking interneurons in the dentate gyrus (defined as "perisomatic-innervating") show both a decrease in peak amplitude and a decrease in decay rate with the distance separating the cells (with the IPSC onset latency used as a surrogate measure of distance). They show by modulating separately the amplitude-distance relationship and the tau-distance relationship that the two have complementary effects on the ability of a simulated network to generate robust local and independent oscillatory assemblies when presented with two spatially-separated 'humps' of excitation. This is interesting and conceptually novel. The authors then attempt to relate the findings to in vitro evoked gamma bursts and show that focal oscillations are more likely to synchronize when they are close together. This is not surprising, and the phenomenon could be achieved simply by restricting the connectivity of a network of inhibitory units in a distance-dependent manner (after all, transecting the tissue would achieve the same phenomenon). Finally, they show that brief bursts of gamma can occur in freely moving animals. Again, this is relatively remote from the simulations. The authors speculate about the number of independent gamma foci that can occur on the basis of the distance-dependent amplitude and tau relationship.

1. The figures are not easy to read because they are extremely small (although carefully prepared, and the manuscript is well written).

We thank the reviewer for praising the quality of our figures and the text. We enlarged the details in our figures following the reviewer's proposal and complying with the guidelines of the journal.

2. The most obvious limitation of the study is that the distance-dependence of connectivity is difficult to infer from brain slices because many axons are cut. The authors ignore this variable and simply assume a Gaussian distribution in the probability of interneurons connecting in their model. As pointed out above, the ability of a network to generate local gamma, and uncouple foci from one another, could be modulated by varying the connectivity. The local connectivity of interneurons in the model is however at variance with the vast majority of the literature on simulated oscillations, where interneurons are assumed to receive and send connections more broadly than principal cells, and therefore are poorly tuned to stimulus.

The connection pattern used in our network model is based on morphological reconstructions of individual basket cells in the DG and CA1, labelled intracellularly in vivo (Sik et al., 1995, J Neurosci 5; Sik et al., 1997, Eur J Neurosci 9). In these studies the spatial distributions of the axon and the synaptic outputs of the labelled basket cell onto target principal cells as well as other PV⁺ INs were quantified. The data show a Gaussian distribution of the synaptic outputs and the axonal length along the transverse axis of the principal cell layer. These published data indicate that PII inhibitory outputs are not broadly and homogeneously distributed in the neuronal network but follow a Gaussian distribution. Our model consists of 200 interneurons (INs) in which a single IN is connected to on average 60 among the 100 nearest neighbor INs and to on average 80 among the 400 nearest principal neuron (PN) neighbors. The connection probabilities of IN-IN and IN-PN synapses were described by a Gaussian distribution with a SD of 25 IN-to-

IN distances (distance between INs 50 μm). In contrast, individual PNs contacted on average 12.5 among the 100 nearest neighbor INs and the SD of the connection probability distribution was only 10 IN-to-IN distances. Indeed, we aimed for a broader axonal distribution of INs than PNs. We emphasize this difference in the mean connectivity and divergence of IN vs PN output in the methods section of our manuscript on page 16.

3. The central proposal that the spatial functional connectivity of PII supports local oscillations for information processing implies that physically neighboring neurons are more likely to participate together in representing information. Although this is the case in motor and sensory cortices, I am not aware of any evidence to support this in the hippocampal formation.

Functional topography has been a topic of intense debate for the last two decades and beyond. There are numerous studies arguing in favor (e.g. Hampson et al., 1999, Nature 402; Nakamura et al., 2010, Neuroscience 166) or against (Redish et al., 2001, J Neurosci 21; Dombeck et al., 2010, Nat Neurosci 11) topographical representations in the hippocampus. Some of these controversies can probably be explained by inconsistencies in the techniques applied for data analysis or the behavioral paradigm used to investigate the spatial organization of cell assemblies encoding information in the network. For example, by applying novel spike sorting algorithms, it has been shown that closely neighboring pyramidal cells in CA1 fire synchronous spikes on the sub-millisecond time scale (Takahashi and Sakurai, 2009, Front Neural Circuits 3:9). Furthermore, in a hippocampus-dependent delayed eyeblink-conditioning task, assemblies of neurons showing high levels of spike correlations were shown to be spatially clustered in CA1 and these spatial clusters underwent changes during the learning task (Modi et al., 2014, eLife 3). In view of these studies and the well-organized topographic projections to and within the hippocampus (Witter, 2007, Prog Brain Res 163; Brivanlou et al., 2004, PNAS 101), it seems that the hippocampus has the potential to form representations preferentially in physically neighboring neurons, depending on the cognitive task it is engaged in. We now included additional references to support this important point in our manuscript (Nakamura et al., 2010; Modi et al., 2014; Brivanlou et al., 2004).

4. In Figure 1 the authors compare the DD situation to the no-DD situation, but this is applied both to PII-PII signaling and to PII-principal cell signaling. They do not separate out the roles of DD versus no-DD for PII-PII specifically. This would require exploring all 4 permutations of DD versus no-DD for PII-PII and PII-principal cells.

We thank the reviewer for pointing to this important question. We have now analyzed, as requested by the reviewer, how the two forms of DD inhibition influence focal neuronal network synchronization. The results are summarized in the new Supplementary Figure 2. We determined the characteristics of focally induced oscillations for different combinations of inhibitory conductance strength and focal stimulation intensities onto PNs. We performed this parameter space exploration in networks with uniform inhibition (noDD), distance-dependent inhibition in either PII-PII synapses or PII-PN synapses, and in the full DD network. Our data show that distance-dependence in PII-PII signaling supports synchronization of focally excited PII networks while distance-dependence in PII-PN synapses improves entrainment of PNs. These two mechanisms jointly boost the power of focally induced gamma activity as shown by the LFP-A measure and enable highly precise spike timing in focally excited principal cell assemblies. These results are now described in the main text in a new section on page 6, 2nd paragraph until page 7, 1st paragraph.

5. Page 5: the gamma frequency is given in some places as kHz. I presume this is the summed firing of many simulated neurons, but what is the average firing rate of each cell?

We thank the reviewer for pointing to this potential source of misinterpretation of the data. These values do not refer to the summed firing of many simulated neurons, but result from a power density analysis of the average momentary firing rates of the simulated neurons. In order to properly address the question of the reviewer, we quantified and added the mean firing rates of PNs and INs during the control and focal stimulation period to the main text (page 5, 2nd paragraph, page 6, 1st paragraph). Furthermore, we added a sentence explaining the unit (kHz) of the firing rate power to the methods section (page 18, 2nd paragraph).

6. Page 5: “significantly weaker fluctuation in its duration...” The supporting statistics do not seem to address the difference in fluctuation, or else they are ambiguous.

We thank the reviewer for spotting this mistake. The given p-value actually refers to the tested difference in variance of the two LFP cycle duration distributions between the noDD and DD neuronal network. We corrected the information on the performed test, which was an Ansari-Bradley test, in the Results section on page 6, 1st paragraph.

Reviewer 2 (Remarks to the Author):

The study by Struber et al. aims to understand the role of distance-dependent inhibition in the generation of focused gamma oscillations in the dentate gyrus using in vitro and in vivo experiments, complemented by computational modeling.

This is an elegant and important study that will likely have significant impact on the field. The results are convincing and novel. The paper is well written and the interpretation of the data is solid. I don't have major concerns; my comments below raise relatively minor points that the authors should be able to address.

1) The authors measure the amplitude and kinetics of PII-PII inhibitory synaptic currents in somatic voltage clamp. As the authors are well aware, cable properties and poor voltage control of dendrites attenuate the amplitude and slow the decay of synaptic currents generated in dendrites and measured at the soma. Therefore, the observed distance-dependence of IPSC amplitude and decay kinetics could be a consequence of a gradient of synapse locations. I.e, if the axons of PII neurons make local synapses preferentially near the soma of proximal PII neurons, and distal synapses preferentially on the dendrites of distal PII neurons, IPSCs recorded at the soma will appear smaller and slower for the dendritic inhibitory events. Do the authors have histological data to address this possible mechanism? If synaptically connected PII interneurons were filled to label dendrite and axon, it would be sufficient to quantify co-localization of axon and dendrite for proximal and distal pairs. In many ways, this is a trivial point, but it should be addressed.

*We thank the reviewer highlighting this important point. The paired recording data were originally acquired in 2002 (Bartos et al., 2002, PNAS 99) and microscopical analysis of the histological specimen is not possible any more. To address the question of the reviewer we performed a new set of experiments in which individual PIIs were intracellularly labelled with biocytin during whole-cell recordings. Antibody labelling against PV was applied to identify and quantify the location and number of putative PII output synapses, characterized as varicosities located in close vicinity of target PV-positive somatodendritic compartments. Our data show that the number of putative contacts declined with distance between pre- and postsynaptic neurons (**Reviewer figure 1, left**), confirming our previous investigation on distance-dependent inhibition at PII-granule cell synapses (Strüber et al., 2015, PNAS 112). Moreover, confocal image stacks show that putative PII-PII synapses were always located at the target soma or the proximal*

apical dendrites, independent of the distance between pre- and postsynaptic PII (**Reviewer figure 1, right**). Finally, the rise time of uIPSCs was not significantly correlated with distance (Spearman correlation analysis: $\rho = 0.496$, $p = 0.0772$). Thus, our data indicate that PII output synapses are located perisomatically at target PII and distance-dependent changes in the amplitude and time course of uIPSCs are unlikely to be explained by attenuation and deceleration of signals at target dendrites.

Reviewer figure 1. Number and mean somatodendritic distance of putative PII-PII synapses on cells located at diverse distances from the presynaptic soma. Individual PII ($N=3$) were filled intracellularly with biocytin in slices of P18 rat dentate gyrus. Biocytin-filled PII and remaining PV⁺ interneurons were fluorescently stained and putative synaptic contacts were identified as close appositions of presynaptic axonal boutons in close vicinity of PV⁺ somatodendritic compartments by confocal microscopy. Left, the number of putative contacts significantly declined with increasing distance between the pre- and postsynaptic soma. Right, the average somatodendritic distance of the identified synaptic contacts remained constant over intersomatic distance. ρ , Spearman's correlation coefficient; p , one-tailed analysis.

2) Nitz & McNaughton, 2004 seems like not a sufficient justification for discarding PN-IN networks in favor of IN-IN networks. That study showed increased firing rates of interneurons during novelty. The paradigm and mechanisms explored in that study are orthogonal to focal gamma generation. The effects on gamma frequency and power of DD vs. noDD in the model are very weak under conditions of physiological levels of feedback excitation (Supp. Fig. 1). I would rather have seen the sensitivity analysis in Fig. 2 for IN-PN networks.

The introduction of feedback excitation moves the conditions of gamma generation from an IN-driven mechanism (ING network) to a more PN-driven mechanism (PING network). We followed the proposal of the reviewer and performed a new detailed quantitative and statistical analysis of focal gamma activity in neuronal network models with PN-driven feedback excitation. These new data are added to the new Fig. 3 (e-j), the former Supplementary Fig. 1. The data show that the power of the LFP-A is substantially higher in DD than in noDD networks under PING conditions (Fig. 3g, left). Due to the enclosure of an additional latency (PN to IN), frequency of gamma oscillations is much lower during conditions of PING than of ING synchronization. However, DD networks which oscillate in the PING regime are always faster than the corresponding noDD networks (Fig. 3g, right). Finally, PN spike coupling to the LFP-A is higher in networks with DD inhibition (Fig. 3h, right). These new sets of data are included in the Results section (page 7, 2nd paragraph). They also show that distance-dependent inhibition at IN-PN synapses plays an important role in the synchronization of neuronal networks that include feedback excitation (PING; Fig. 3j), but distance-dependent inhibition at IN-IN synapses is critical for the synchronization of neuronal networks based on ING mechanisms. Since in our recent study (Strüber et al., 2015, PNAS 112), we have already examined the mechanisms of PN synchronization by DD inhibition at PII-PN synapses, our experimental results on PII-PII synapses prompted us to identify the contribution of DD in the amplitude and the time course of inhibition to focal ING oscillations. We therefore focused the remaining computational analysis in the manuscript on the role of DD inhibition in synchronizing IN networks. This focus reduced the computational complexity and thereby improved the conclusions of our model. We hope that the reviewer can support our argumentation.

3) In the discussion it is stated that high frequency gamma “is independent of AMPA receptor-mediated excitation but requires fast GABAAR-driven signaling.” The evidence for this seems rather weak, based on bath application of DNQX (Jackson et al., J. Neurosci., 2011). DNQX and CNQX are partial agonists of neuronal AMPARs that directly depolarize interneurons (Maccaferri & Dingledine, Neuropharmacology, 2002; Menuz et al., Science, 2007). If this is an important point for the study, the authors could test if NBQX, a true competitive antagonist of AMPARs, blocks gamma in their slice model.

Following the proposal of the reviewer we induced gamma oscillations in slice preparations by puff application of high K⁺ solution to the molecular layer in control conditions and in the presence of 20 μM NBQX in the bath. This manipulation resulted only in a mild reduction of the amplitude of gamma oscillations by 36% (6 slices, 2 rats), but did not abolish the network’s capability to generate gamma in the same frequency range. This reduction in amplitude is expected because glutamatergic signaling activates neurons and supports the emergence of rhythmic activity patterns. Our data markedly contrast previous investigations in CA1 and CA3 showing that 20 μM NBQX fully abolished carbachol induced gamma rhythms, a known PING oscillation model (Fisahn et al., 1998, Nature 394). Thus, under our experimental conditions AMPA receptor mediated excitation is not needed for the emergence of gamma oscillations in the dentate gyrus. Our data further suggest that interneurons in our K⁺ puff application model are recruited additionally by other mechanisms, very likely direct depolarization. In contrast to AMPA receptor mediated synaptic transmission, GABAA receptors seem to be necessary for gamma generation, as in one experiment bath-application of the GABAA receptor blocker SR95531 (10 μM) fully abolished gamma activity patterns (reduction of power in the 25-45 Hz range by 77%). These results are consistent with previous data in the dentate gyrus applying the same method for inducing in vitro oscillations and suggest that an ING mechanism underlies the observed gamma activities (Towers et al., 2002, J Neurophysiol. 87). These data are included in the new Supplementary Fig. 7.

4) In the previous study (Struber et al., 2015), the α1-subunit-selective GABAAR agonist zolpidem was used to normalize the decay kinetics of proximal and distal IPSCs. What does zolpidem do to gamma frequency and power in the slice model?

We performed the requested experiments: Similar to the experimental design described in our response to Point 3, we induced gamma oscillations in acute slice preparations of the dentate gyrus and bath-applied 1 μM zolpidem (8 slices, 1 rat). We observed a significant reduction to $52.3 \pm 9.7\%$ of the oscillatory amplitude with no significant changes in the oscillatory frequency. These data support previous in vitro studies showing that bath-application of diazepam, which increases the time course and amplitude of GABAA receptor-mediated conductances, strongly reduces the synchrony but only mildly affects gamma frequency (Traub et al., 1996, J Physiol Lond 493). These data have been added to the new Supplementary Fig. 7.

5) I appreciate that it is no trivial endeavor to combine in vivo, in vitro, and in silico experiments. However, the flow of the current manuscript is a bit disjointed. Ideally, the paper would start with the observation of focal gamma in vivo, exploit the slice model of focal gamma to tease apart mechanisms, and use the model to make experimentally testable predictions that could be confirmed either by new analysis or perturbation of the experimental system.

We thank the reviewer for her/his suggestion on the structure of the manuscript. We re-ordered the sequence of the described experiments in the Results section as requested and start now with the in vivo observation of focal gamma activity. We hope the reviewer can agree with our impression that comprehensibility of the presented findings has gained substantially by these changes.

Reviewer 3 (Remarks to the Author):

In vivo recordings from primate visual cortex have established that the frequency and spatial coherence of gamma-frequency oscillations are stimulus-dependent. The functional significance of such changes in gamma synchronization continues to be debated, but it is clear that our fundamental understanding of how cortical circuits operate must include an explanation of such activity-dependent changes in dynamic coupling.

In this manuscript, Struber et al show that local and global patterns of gamma synchronization also occur in the rodent dentate gyrus in vivo. Using paired recordings from perisomatic-targeting interneurons and extensive computational modelling, they suggest that distance-dependent properties of GABAergic synapses enable/enhance the activity-dependent spatial divergence of gamma synchronization.

It is well-established that conduction delays, inhibitory conductance and synaptic decay constants influence the spatiotemporal patterns of fast network oscillations (eg Brunel & Wang, 2003; PMID:12611969; plus the authors previous work). In a recent modelling study of the visual cortex, it has also been suggested that the frequency and spatial spread of gamma oscillations depends on both the difference in drive and coupling strength between local circuits (Lowet et al, 2015; PMID:25679780). However, the role of distance-dependent of GABAergic signalling may be an important new concept, particularly for explaining focal gamma oscillations in inhibitory networks.

Main concerns:

1) The authors show that the peak amplitudes and decay time constants of uIPSC vary with axonal distance (Fig 1b), which is contrary to the assumption that distal synapses would be less likely to occur, but have similar properties. To test the effects of 'distance-dependent inhibition' (DD) vs noDD in network models, the conductance integral is kept constant. This means that the amplitude is always larger and/or decay time constant always faster for local connections in the DD vs noDD model. In the case of Fig 1d, this means for the DD network there is strong fast inhibition at the centre of stimulation, and slow lateral inhibition (completing shutting down neighbouring neurons; removes interference from conduction delays and variance in synaptic decay constants), which has understandable effects on frequency and synchronization (eg Brunel & Wang, 2003).

It is reasonable to examine the effects under conserved average inhibitory synaptic strength, and the authors may feel that they have actually demonstrated whether this interpretation is correct or not. However, I think there should also be comparisons with DD and noDD networks that have the same absolute local amplitudes/decay time constants, which I believe was the null hypothesis for the uIPSC data. This will clearly have effects on baseline and/or stimulus-induced activity, but can be compensated for by changes in tonic GABA conductance / excitatory drive (rather than parameters of synaptic inhibition), and could establish the importance local vs peripheral parameters in the balanced model presented.

We thank the reviewer for pointing to the central question of how to model DD inhibition. We were indeed considering different strategies of representing DD inhibition in a comparison with uniform inhibition. Previous publications including our own (e.g. Vida et al., 2006, Neuron 49) showed that strength of synaptic inhibition has a strong influence on neuronal network synchronization. We therefore decided to adjust the conductance integral in both network models (noDD, DD) to create a 'balanced' DD network (integral was the same for both models) and to exclude possible side effects caused by differences in the total inhibitory strength. However, we agree that from the perspective of a synaptic electrophysiologist it will be of interest to compare DD and noDD networks that have the same absolute local amplitudes/decay time constants. We therefore performed a new set of network simulations in DD and noDD networks with equal inhibitory amplitudes and kinetics for closest neighbors. As hypothesized, due to the lower compound inhibition, baseline firing rates were higher in this 'unbalanced' DD network (INs: noDD 36.05 ± 1.80 Hz, DD balanced

28.31 ± 1.47 Hz, DD unbalanced 43.04 ± 1.60 Hz; PNs: noDD 0 Hz, DD balanced 0.02 ± 0.016 Hz, DD unbalanced 0.05 ± 0.047 Hz). Interestingly, after focal stimulation we observed that the LFP-A power was smaller in the unbalanced DD than in the balanced DD network model, but higher than in the noDD network. These data indicate that IN activity was still more synchronous in the unbalanced DD than in the noDD network (Supplementary Fig. 1b). We further confirmed this finding by comparing the coupling of IN activity to the LFP-A. Higher IN synchrony and consequently more powerful output inhibition in the unbalanced as well as in the balanced DD network was very effective in controlling spike timing in principal neurons, quantified by the coupling of PN activity to the LFP-A. Thus, the distribution of distance-dependent synaptic properties and not their actual values seem to be the crucial factors supporting a high power of focal gamma oscillations and a tight control of spike timing in principal neurons. These data are now shown in the new Supplementary Fig. 1.

2) Fig 4 - The noDD model suggests that distance does not have a major effect on gamma cross-correlations for 2 equal stimuli separated by up to 800 micron. This fits with the connectivity probability shown in Fig 1c (along with x axis of Fig 1b; -> ~100%) – how was this determined for the model?

The topology of the PII network is based on anatomical data obtained by Sik et al. (1995, J Neurosci 15) for the CA1 basket cell network. These data have been first used by Wang & Buzsáki (1996, J Neurosci 16) in order to construct an anatomically realistic network model of hippocampal interneuron circuits. This model has since been used frequently for studies of interneuron function in neuronal network models (e.g. Bartos et al., 2002, PNAS 99; Vida et al., 2006, Neuron 49). We therefore adapted these parameters and incorporated our distance-dependent concept of inhibition in this well-established neuronal network model. See also response to reviewer #1 point 2.

Whatever the starting conditions, how changes in local and peripheral inhibition affect the localization of gamma activity seems readily interpretable. However, I do not understand the experimental manipulations / analysis:

- i) What is the spatial resolution of the K⁺ pressure ejection, and what is the activity induced at each site alone
- ii) What is the spatial resolution of the LFP recording (, and what is the activity induced at each site alone)
- iii) At 700 microns is LFP2 suppressed by site 1 stimulation? – this is the example recording, but the LFP recordings look very different in amplitude, which is not analyzed. What was the response to single site stimulation? How does this relate to the Stimulus 2/ Stimulus 1 ratio in Fig 4a?
- iv) I assume the cross-correlation coefficient is taken at lag 0. This must be higher at closer spaced LFP electrodes. However, the side bands look more prominent for wider spaced stim/LFP electrodes, which would be more indicative of rhythmic synchronization.

We thank the reviewer for addressing these important questions related to the in vitro oscillation model. We performed additional experiments to reply to the following points raised by the reviewer.

(i-iii) Our data analysis shows that the oscillations induced by single site stimulation do not substantially differ in their power from paired stimulations (Supplementary Fig. 7a). These results argue against a direct suppressive effect of one stimulation site on the evoked oscillations at the other stimulation site. Furthermore, we quantified the spatial profile of the K⁺ puff-evoked in vitro oscillations and confirmed their very focal nature (Supplementary Fig. 7d).

(iv) Fig. 6d (the former Fig. 4) depicts the maximum values in the cross-correlograms. Thus, the side bands mentioned by the reviewer are actually taken into account in our quantification. Indeed, in our analysis we do not want to exclude any synchronization between both oscillatory foci. However, we show that the extent of synchronization decreases with distance between both foci, similar to the DD network model. Actually, from our simulations we would predict that transient synchronization between both stimulation sites is possible under distinct conditions, namely, if the amount of excitation and, in consequence, the evoked

oscillation frequencies are in a similar range (cf. the DD model for equal stimulation strengths, see Fig. 6b lower plot, for 100% amplitude stimulus 2; similar principles are described in Lowet et al., 2015, PLoS Comput Biol. 11).

Other points:

1) The firing rates in the models of interneurons and principal cells are very high, with each neuron likely showing strong gamma side peaks in the spike time autocorrelations even under baseline conditions. Page 4, line 2 – ‘principal cell activity is extremely sparse’ – this is not reflected in the model (Fig 1). The activity of the interneurons is not representative of in vivo spike patterns either. Is it possible for DD inhibition to be effective with sparse firing?

The focal stimulation in our model represents a transient cortical input that conveys salient sensory information to the hippocampal network but not baseline activity levels as usually reported in in vivo studies. The situation described by the quoted sentence ‘principal cell activity is extremely sparse’ is represented by the control period in our simulations, where INs in the DD network are active at 28.31 ± 1.47 Hz, while PNs are almost silent with an average activity of 0.02 ± 0.016 Hz. This activity fits well, for example, to the average firing frequency of 25 ± 8.4 Hz of CA1 PV⁺ INs during running episodes (Varga et al., 2012, PNAS 109). During the stimulation paradigm, activity levels might increase substantially but only for short periods of time. Such a drastic increase in activity is possible as shown for PV⁺ INs which increase their discharges from 8.2 ± 5.6 Hz during baseline conditions to 75 ± 17 Hz during ripple oscillations (Varga et al., 2012 PNAS 109). In contrast, PNs have been shown to be only sparsely active during baseline conditions, in particular in the dentate gyrus (mean firing frequency of 0.5 Hz in active GCs, Pernia-Andrade and Jonas, 2014, Neuron 81), but can increase transiently their activity to high firing frequencies in response to stimuli according to the cells’ receptive fields (e.g. peak activity in the place fields in the hippocampus; intra-burst firing frequencies of >100 Hz; Pernia-Andrade and Jonas, 2014, Neuron 81). However, we would like to emphasize that distance-dependent inhibition is also highly effective in controlling PN spike timing for lower PN firing rates (see our new Supplementary Fig. 2d and f).

It should be noted that the method of analyzing coherence based on pairwise comparisons depends on spike rate (are they active every cycle of a network rhythm), as well as synchrony. This is covered by the additional analyses of LFP-A and spike coupling in Fig 1, but these additional analyses are not always presented for parameter variation, and may be more important measures with low spike rates.

We thank the reviewer for this very important comment, which we addressed as requested. Indeed, pairwise correlation measures heavily depend on the joint firing rate of two neurons (e.g. Dorn and Ringach, 2003, J Neurophysiol 89) and the question underlying the reviewer’s comment is whether differences in firing probability between the DD and the noDD network can explain the observed differences in the coherence (e.g. Fig. 4a). By employing a variable time bin size that depends on the dominant oscillation frequency for calculating the coherence measure kappa, we already reduced the strong influence of the firing rate on coherence. However, to fully address this question, we conducted two additional analyses. Firstly, for the IN-PN network model we always analyzed the spike coupling of INs and PNs to the simultaneously obtained LFP-A. Secondly, for the IN network model, we calculated for all four network conditions (noDD, DDamp, DD τ , full DD) the ‘spike time tiling coefficient’ (STTC; Cutts and Eglon, 2014, J Neurosci 34) to quantify IN synchrony. This alternative spike correlation measure is independent of firing rates. The results are summarized in the new Supplementary Fig. 4 and show that the identified effects of distance-dependent inhibition on the synchrony of IN discharges in the model cannot be explained by differences in their activity levels.

2) Figure 2a, left panel– plots of cell index and coherence do not appear to be vertically aligned.

We rechecked the alignment of all plots in this figure and, where required, readjusted them as requested.

3) The details of the electrophysiology experiments do not appear to be provided.

The methods section describes the applications used for in vivo and in vitro recordings of neuronal network oscillations in the dentate gyrus. The paired recording data shown in Fig. 2a,b originate from previously published whole-cell patch-clamp recordings in acute slice preparations (Bartos et al. 2002, PNAS 99) as highlighted on page 4, 2nd paragraph. In response to the reviewer's request, we now included a new paragraph in the methods section, describing briefly the recording conditions and referring to this publication (page 15, last paragraph).

4) Supplementary Fig 1 – I do not see consistent effects of the DD vs noDD network with E-I connections, and it is not clear this has been analysed statistically. I would assume that E-I connections should be able to break down any effects of DD inhibition, at least to some extent, but this not preclude it from being an important factor in the operation of inhibitory networks.

Please also see response to comment 2 of reviewer #2. To elaborate more on the role of distance-dependent inhibition in network synchronization by ING versus PING mechanisms, we included a new detailed quantitative and statistical analysis of the focal gamma activity in networks with different synaptic strengths of the PN-IN feedback excitation. Our new set of computational data shows that the power of the LFP-A is substantially higher in the DD than in the noDD network model under PING conditions including feedback excitation onto INs. Furthermore, the frequency of gamma oscillations is much lower in networks dominated by a PING than an ING mechanism. Please note that DD networks oscillate in the PING regime faster than the corresponding noDD networks. Finally, PN coupling to the LFP-A is higher in networks with DD inhibition. Taking the comments of reviewers #2 and #3 on board, we decided to emphasize the role of DD inhibition in PING oscillations and moved these new results to the main body of the manuscript. The resulting new set of data resulted in the new Fig. 3, the former Supplementary Fig. 1.

REVIEWERS' COMMENTS:

Reviewer #1 (Remarks to the Author):

The authors have re-organized the manuscript and included additional simulations. The central claim regarding the effect of distance dependence on the ability of a network to sustain spatially segregated gamma oscillations, is interesting. There are nevertheless some inevitable gaps between the experimental data and the simulations. For instance, as pointed out previously, interneurons are generally poorly tuned to inputs when compared to principal cells, whilst the simulations in the manuscript assume that they receive the same "hump" of afferent excitation as the principal cells. The authors mainly concentrate on "ING", but this assumption may be a weak point in extrapolating to "PING".

Reviewer #2 (Remarks to the Author):

The authors addressed all of my concerns. I have no additional questions.

Reviewer #3 (Remarks to the Author):

The authors have performed more experiments and simulations, which significantly strengthen their conclusions on the role of distance-dependent inhibition in the generation of gamma oscillations. This additional data has also addressed all my concerns.

I only note 2 issues related to the pharmacology experiments presented in Supplementary Fig 7:

- i) NBQX should be described as an AMPA/kainate receptor blocker
- ii) The rationale and results for these experiments should be highlighted briefly in the main text.

Point-by-point response to the reviewers' comments:

We thank the Associate editor and the reviewers for appreciating the major amendments we have made to improve our manuscript in response to the editor's and the reviewers' comments. Two reviewers requested some changes, which we addressed and reply to in the following point-by-point response:

Reviewer #1 (Remarks to the Author):

The authors have re-organized the manuscript and included additional simulations. The central claim regarding the effect of distance dependence on the ability of a network to sustain spatially segregated gamma oscillations, is interesting. There are nevertheless some inevitable gaps between the experimental data and the simulations. For instance, as pointed out previously, interneurons are generally poorly tuned to inputs when compared to principal cells, whilst the simulations in the manuscript assume that they receive the same "hump" of afferent excitation as the principal cells. The authors mainly concentrate on "ING", but this assumption may be a weak point in extrapolating to "PING".

We thank the reviewer for highlighting the importance of the central claim of our study. We fully agree that, in general, cortical interneurons seem to be more poorly tuned to excitatory inputs than principal cells. Indeed, this was the main reason for targeting the focal excitatory input onto principal cells but not interneurons in our PING model of the revised manuscript. The "hump"-like activity of interneurons in these simulations is inherited from feedback excitation provided by the locally active principal cells, as it was already suggested e.g. by Nitz & McNaughton (J Neurophysiol 91, 2004), and is not generated by the external excitatory drive. In the Discussion on page 13, end of the 2nd paragraph of our revised manuscript, we explain the specifics of the excitatory input required to evoke focal ING and PING oscillations in the network. We hope that by including this additional information we improved the clarity of our simulations.

Reviewer #3 (Remarks to the Author):

The authors have performed more experiments and simulations, which significantly strengthen their conclusions on the role of distance-dependent inhibition in the generation of gamma oscillations. This additional data has also addressed all my concerns. I only note 2 issues related to the pharmacology experiments presented in Supplementary Fig 7:

- i) NBQX should be described as an AMPA/kainate receptor blocker
- ii) The rationale and results for these experiments should be highlighted briefly in the main text.

We thank the reviewer for praising the new experiments and simulations we performed to address the comments of the reviewers and which helped us to improve the quality of our manuscript. We corrected the description of NBQX as an AMPA/kainate receptor blocker in the legend of Supplementary Fig. 8. We, furthermore, included a new section in the Results part of our manuscript, in which we briefly explain the rationale and the results of the new pharmacological in vitro oscillation experiments (page 12, 1st paragraph).